# AUTOG: TOWARDS AUTOMATIC GRAPH CONSTRUCTION FROM TABULAR DATA

**Zhikai Chen**[1,*] **Han Xie**[2], **Jian Zhang**[2], **Xiang Song**[2], **Jiliang Tang**[1],
**Huzefa Rangwala**[2], **George Karypis**[2]
[1]Michigan State University    [2]Amazon

## ABSTRACT

Recent years have witnessed significant advancements in graph machine learning (GML), with its applications spanning numerous domains. However, the focus of GML has predominantly been on developing powerful models, often overlooking a crucial initial step: constructing suitable graphs from common data formats, such as tabular data. This construction process is fundamental to applying graph-based models, yet it remains largely understudied and lacks formalization. Our research aims to address this gap by formalizing the graph construction problem and proposing an effective solution. We identify two critical challenges to achieve this goal: 1. The absence of dedicated datasets to formalize and evaluate the effectiveness of graph construction methods, and 2. Existing automatic construction methods can only be applied to some specific cases, while tedious human engineering is required to generate high-quality graphs. To tackle these challenges, we present a two-fold contribution. First, we introduce a set of datasets to formalize and evaluate graph construction methods. Second, we propose an LLM-based solution, AutoG, automatically generating high-quality graph schemas without human intervention. The experimental results demonstrate that the quality of constructed graphs is critical to downstream task performance, and AutoG can generate high-quality graphs that rival those produced by human experts. Our code can be accessible from `https://github.com/amazon-science/Automatic-Table-to-Graph-Generation`.

## 1 INTRODUCTION

Graph machine learning (GML) has attracted massive attention due to its wide application in diverse fields such as life science (Wong et al., 2023), E-commerce (Ying et al., 2018), and social networks (Wang & Kleinberg, 2023; Suárez-Varela et al., 2022). GML typically involves applying models like graph neural networks (GNNs) (Kipf & Welling, 2017; Ma & Tang, 2021) to leverage the underlying graph structure of a given task, e.g., using the friendship networks for user recommendations (Tang et al., 2013) and identifying new drug interactions (Zitnik et al., 2018).

Despite the widespread interest and rapid development in GML (Kipf & Welling, 2017; Ma & Tang, 2021), constructing graphs from common data formats such as industrial tabular data (Ghosh et al., 2018) remains an under-explored topic. This primarily stems from a widely adopted assumption that appropriate graph datasets exist for downstream tasks akin to established benchmarks (Hu et al., 2020a; Khatua et al., 2023). However, readily available graph datasets are absent in many real-world enterprise scenarios. First, given an input data in common storage formats such as tables, there can be many plausible graph schemas and structures that can be defined over them. The choice of graph schema impacts downstream performance of GML. Rossi et al. (2024) shows that considering the directional aspect of edges within a graph can lead to substantial variance in the downstream GML performance. Second, converting the source data into graph format requires expert data engineering and processing. Even though, GNN based approaches shows strong performance on Kaggle leaderboard (Wang et al., 2024c), it involves laborious pre-processing and specialized skills to transform the original tabular data into ready-to-be-consumed graphs for GML.

---

*The work is done while being an intern at Amazon.

The objective of this work is to formalize the challenges in graph construction by establishing real-world datasets followed by automatic graph construction from input tabular data.

Existing tabular graph datasets such as Wang et al. (2024c) and Fey et al. (2024) assume the availability of well-formatted graphs with explicit relationships such as complete foreign-key and primary-key pairs. In these cases, graphs can be easily constructed using heuristics like Row2Node (Cvitkovic, 2020) by converting each table into a node type. However, implicit relationships like columns with similar semantics (Dong et al., 2023) or columns with categorical types also widely exist in real-world scenarios, which cannot be addressed by heuristic methods (see Figure 1). Datasets designed for evaluating graph construction should reflect the importance of modeling implicit relationships. Additionally, different tasks can be defined based on the same dataset (Fey et al., 2024). Further, different ways to construct graphs affect different tasks' performance is an understudied problem. Therefore, ideal datasets for graph construction should also include different downstream tasks to reflect this problem. *From the solution perspective*, graph construction involves finding the best candidate among all possible graph structures. However, considering the vast search space, finding the graph structure through an exhaustive search is infeasible. Therefore, an effective automatic graph construction method should be able to efficiently identify high-quality candidates from many possible graph structures/schemas.

To address the above challenges, we propose a set of evaluation datasets and a large language model (LLM)-based graph construction solution. We first extract raw tabular datasets from Kaggle, Codalab, and other data sources to design datasets reflecting real-world graph construction challenges. They differ from prior work (Fey et al., 2024; Wang et al., 2024c) in that these datasets haven't been processed by experts, and existing graph construction methods get inferior performance (see Table 3). To solve the graph construction problem, we propose an LLM-based automatic graph construction solution **AutoG** inspired by LLM's reasoning capability to serve as a planning module for agentic tasks (Zhou et al., 2024) and tabular data processing (Hollmann et al., 2023). However, we observe that LLMs tend to generate invalid graphs or graphs with fewer relationships (as shown in Section 5.3.1). We address this problem by guiding LLMs to conduct close-ended function calling (Schick et al., 2024). Specifically, we decompose the generation of graph structures into four basic transformations applied to tabular data: (1) establishing key relations between two columns (finding cross-table relationships), (2) expanding a specific column (finding self-induced relationships), (3) generating new tables based on columns (data normalization), and (4) manipulating primary keys (generating proper node and edge types). Coupled with chain-of-thought prompt demonstrations for each action, AutoG generates a series of actions to get the augmented schema and thus construct the graph. We further demonstrate the effectiveness of AutoG on our proposed benchmarks.

Our major contributions can be summarized as follows:

a) **Formalizing graph construction problems with a set of datasets**: We create a set of datasets covering diverse graph construction challenges, consisting of eight datasets from academic, E-commerce, medical, and other domains.

b) **LLM-based automatic graph construction method: AutoG**: To solve the graph construction problem without manual data engineering, we propose an LLM-based baseline to automatically generate graph candidates and then select the best candidates efficiently.

c) **Comprehensive evaluation**: We compare AutoG with different baseline methods on the proposed datasets. AutoG shows promising performance that is close to the level of a data engineering expert.

## 2 PRELIMINARIES

### 2.1 TABULAR DATA AND SCHEMAS

The input tabular data is represented using the RDB language (Codd, 2007; Chen, 1976) as a schema file. Subsequently, we introduce table schemas and how they may be used to describe a graph. We start by introducing the fundamental elements of RDB languages.

**Definition.** Tabular data $\mathcal{D}$ contains an array of $K$ tables $\mathcal{D} := \{T_i\}_{i=1}^{K}$. Each table $T_i$ can be viewed as a set $T_i = (C_i, R_i, M_i)$, where

- $C_i = (C_{i,1}, \ldots, C_{i,l_i})$ is an array of strings representing the column names, with $l_i$ denoting the number of columns in $T_i$.
- $R_i$ is a matrix where each row $R_{i,j} = (R_{i,j,1}, \ldots, R_{i,j,l_i})$ contains the values for the $j$-th row of table $T_i$.
- $M_i = (M_{i,1}, \ldots, M_{i,l_i})$ is an array specifying the data type of each column.

In this paper, we consider the following data types {`category`, `numeric`, `text`, `primary_key`(PK), `foreign_key` (FK), `set`, `timestamp`}. As an example, if $M_{i,1} = $ `text`, then all values in the same column $R_{i,1,1}, \cdots, R_{i,m_i,1}$ are of text type ($m_i$ refers to the number of rows for table $T_i$). Detailed descriptions of each data type can be found in Appendix A.1.

The definitions above focus on the properties of individual tables, For multiple tables with $K > 1$, they can be related with set of $n$ `PK-FK` pairs $\{x_{\text{PK}}^m, y_{\text{PK}}^m, x_{\text{FK}}^m, y_{\text{FK}}^m\}$ where $m = 1, \ldots, M$. $x$ and $y$ represent the indices of tables in $\mathcal{D}$ and the indices of columns. In this paper, we consider the scenario that multiple tables are retrieved for a specific downstream task and no explicit key relationships are given (Gan et al., 2024).

**Table schema and graph schema description.** Based on this language, we define table schema by storing all the meta information in a structured format like YAML (Ben-Kiki et al., 2009). An example is shown in Appendix A.2. Table schema defines the metainformation of tables in a structured manner following the RDB language. Graph schema is a special type of table schema. Compared to general table schema, graph schema presents tables with proper column designs and `PK-FK` relations. These characteristics make it trivial to convert a graph schema (as discussed in Section 2.2) into an ideal graph structure for downstream tasks.

## 2.2 Bridging tabular data and graphs

Based on the definition of tabular data, the goal of graph construction is to convert relational tabular data $\mathcal{D}$ into a graph $\mathcal{G}$. Following Fey et al. (2024); Wang et al. (2024c), we consider $\mathcal{G}$ as a heterogeneous graph (Wang et al., 2022) $\mathcal{G} = \{\mathcal{V}, \mathcal{E}\}$ characterized by sets of nodes $\mathcal{V}$ and edges $\mathcal{E}$. The nodes and edges are organized such that $\mathcal{V} = \bigcup_{v \in V} \mathcal{V}^v$ and $\mathcal{E} = \bigcup_{e \in E} \mathcal{E}^e$ where $\mathcal{V}^v$ represents the set of nodes of type $v$, and $\mathcal{E}^e$ represents the set of edges of type $e$. The main challenge of graph construction lies in extracting appropriate node types and edge types from the schema of tabular data. This process could be straightforward if we treat each table as a node type and each `PK-FK` relationship as an edge type. However, this method may generate suboptimal graphs for general table schemas. For instance, when two entities are placed in a single table, one entity might be treated as a feature of the other, resulting in a graph that fails to effectively reflect structural relationships, thereby impacting the performance of downstream tasks (Wang et al., 2024c).

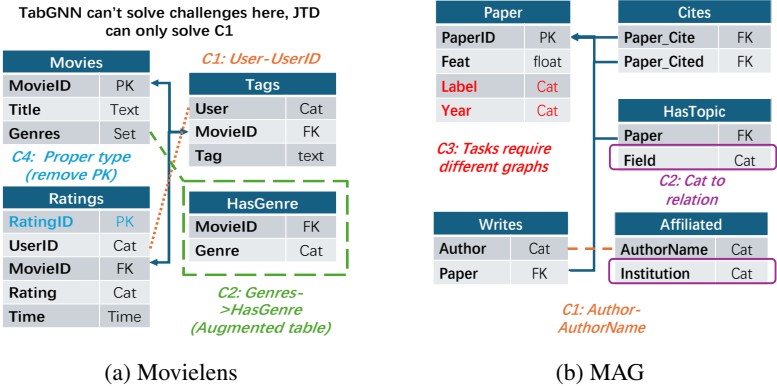

(a) Movielens    (b) MAG

Figure 1: Demonstrations of challenges in two selected datasets. Existing heuristic-based methods cannot well tackle C2-C4 in that they require task-specific decisions.

## 3 Designing datasets for evaluating graph construction

To make the graph construction problem concrete and provide a set of datasets for comparing different methods, we first identify key problems that need to be addressed during the graph construction

process. Based on these problems, we have carefully selected 8 multi-tabular datasets from diverse domains to construct datasets for graph construction.

## 3.1 DESIGN SPACE OF THE DATASETS

We first propose four core challenges to be addressed when converting tabular data into graphs. Examples of these challenges are demonstrated in Figure 1.

1. **C1: Identifying missing cross-table `PK-FK`/`FK-FK` relationships**: Traditional methods like Row2Node (Wang et al., 2024c) only turn `PK-FK` relationships into edges, while these relationships are usually not complete, which necessitates either automatic join discovery (Dong et al., 2023) or human intervention. Compared to traditional join discovery whose aim is to merge tables together, the role of identifying cross-table `PK-FK` is to find proper edge relationships beneficial to the downstream tasks, which is more challenging.
2. **C2: Identifying self-induced relationships**: Beyond the relationship across different tables, tables may present relational connections with their columns. For example, the "Field" column in Figure 1 can induce useful relations, and thus, an augmented table should be added. This corresponds to the single FK problem discussed in Gan et al. (2024). Identifying such self-induced relationships has been shown to be beneficial for tasks like recommendation (Wu et al., 2021).
3. **C3: Transforming tables into proper node or edge types**: How to convert tables into appropriate types affects downstream task performance and the validity of generated graphs. For instance, the "Writes" table in Figure 1 should be better modeled as an edge type since it contains two foreign keys, and records the citation relationship between papers in table "Paper".
4. **C4: Generating proper graphs for different downstream tasks(*)**: Considering that multiple tasks can be defined based on the same tabular data (Fey et al., 2024), one single graph design may not fit all tasks. This claim has not been well studied and will be verified in the experiment. This task is the most challenging and may require substantial work.

**Design philosophy of these challenges.** These four challenges are inspired by existing works (Wang et al., 2024c; Dong et al., 2023; Gan et al., 2024) but go beyond their scopes. Specifically, **C1** is a common problem in data lakes and RDB (Dong et al., 2023; Hulsebos et al., 2019) for automatic data engineering. When constructing the graph is the final objective, joinable column detection becomes even more important since it's crucial to find relations. **C2** is derived by comparing the original schema from Kaggle to the graph schema used in Wang et al. (2024c). Human experts have introduced multiple augmented tables, which are crucial to the performance of GML models. The mechanism behind these augmented tables hasn't been well studied, and we first introduce them in our datasets. **C3** is derived from real-world datasets such as (Harper & Konstan, 2015), and we find that simple heuristics may work poorly when the proper type of table cannot be induced from the schema. **C4** is naturally derived from the multiple tasks defined on tabular data. We are the first to study the influence of graphs on different downstream task performance.

**Relationship to traditional database profiling (Abedjan et al., 2015)**. Database profiling, including normalization, is a related concept to our work. The goal of graph construction from relational data to graph is to find what kind of relational information is beneficial to the downstream task. For example, the objective of challenge 2 is to consider whether the relationship induced by this categorical value is beneficial. This decision needs to consider the semantic relationship between this column and the corresponding downstream tasks, which cannot be solved by normalization. As a comparison, profiling aims to minimize data redundancy and improve data integrity. Despite the overlap, data profiling method cannot fully solve the graph construction task. In Section 4, we also design two functions which can conduct data normalization, and we find that LLM is itself a plausible decision maker for data normalization.

## 3.2 DATASETS

Based on the design space of graph construction from relational tabular data, we gather 8 datasets from various domains to evaluate graph construction methods. We collect these datasets from 1. the source of existing tabular graph datasets, such as Outbrain (Wang et al., 2024c); 2. augmented from existing tabular graph datasets, such as Stackexchange (Wang et al., 2024c); 3. traditional tabular datasets adapted for graph construction, including IEEE-CIS (Howard et al., 2019) and Movie-

lens (Harper & Konstan, 2015). The information of these 8 datasets is listed in Table 1. Two concrete examples are shown in Figure 1. Details on dataset sources and pre-processing are shown in Appendix B.

Table 1: Our proposed datasets. The tasks are categorized into predictions of relation attribute, entity attribute, and FK by following (Wang et al., 2024c).

| Name of the dataset | #Tasks | #Tables | Inductive | C1 | C2 | C3 | C4 | Task type | Source of datasets |
|---|---|---|---|---|---|---|---|---|---|
| Movielens | 1 | 3 | ✓ | ✓ | ✓ | ✓ | ✗ | Relation Attribute | Designed from Harper & Konstan (2015) |
| MAG | 3 | 5 | ✗ | ✓ | ✓ | ✓ | ✓ | Entity Attribute, FK Prediction | Augmented from Wang et al. (2024c) |
| AVS | 2 | 3 | ✓ | ✓ | ✓ | ✓ | ✓ | Entity Attribute | Augmented from Wang et al. (2024c) |
| IEEE-CIS | 1 | 2 | ✗ | ✗ | ✓ | ✓ | ✗ | Entity Attribute | Designed from Howard et al. (2019) |
| Outbrain | 1 | 8 | ✓ | ✓ | ✓ | ✓ | ✗ | Relation Attribute | Augmented from Wang et al. (2024c) |
| Dignetica | 2 | 8 | ✓ | ✓ | ✓ | ✓ | ✓ | Relation Attribute, FK Prediction | Augmented from Wang et al. (2024c) |
| RetailRocket | 1 | 5 | ✓ | ✓ | ✓ | ✓ | ✗ | Relation Attribute | Augmented from Wang et al. (2024c) |
| Stackexchange | 3 | 7 | ✓ | ✓ | ✓ | ✓ | ✓ | Entity Attribute | Augmented from Wang et al. (2024c) |

# 4 METHOD

This section introduces an automatic graph construction solution to tackle the four challenges in Section 3.1. As discussed in Section 2.2, we consider graph construction as a transformation from the original table schema with implicit relations to the final graph schema with explicit relations. We adopt an LLM as the decision maker to generate transformations automatically.

## 4.1 AUTOG: AN LLM-BASED GRAPH CONSTRUCTION FRAMEWORK

In previous work (Fey et al., 2024; Wang et al., 2024c), human data scientists often play the generator, which generates outputs based on their expert knowledge. Like humans, LLMs also demonstrate the capabilities to generate molecular structures or code-formatted augmentations based on prior knowledge (Wang et al., 2024a; Hollmann et al., 2023). Consequently, we adopt an LLM as a generator and provide it with input tabular data to generate transformations.

Following the "pre-processing, modeling, and post-processing" pipeline of common data science engineering (Biswas et al., 2022), we propose an automatic graph construction framework consisting of three modules: (1) Input context module, which provides essential metadata for the LLM to make decisions; (2) Generator module, which generates the augmented data together with updated data tables based on the provided context; (3) Discriminator module, which evaluates the generated schema and output feedback. The whole pipeline is demonstrated in Figure 2.

**Input context module.** The input context module is designed to equip AutoG with essential insights about the input data. Given the assumption that AutoG processes typeless tabular data, this module first performs a preliminary analysis by extracting key statistics such as the number of samples, unique samples, and mode values. These statistics enable LLMs to distinguish between categorical and numerical columns, critical for downstream edge relationship discovery. Accurately inferring column data types can significantly influence downstream task performance. For instance, a text-valued column might be interpreted as either text (encoded as dense text embeddings) or categorical (represented via learnable embeddings or transformed into self-induced relationships). The prompt for column type inference is de-

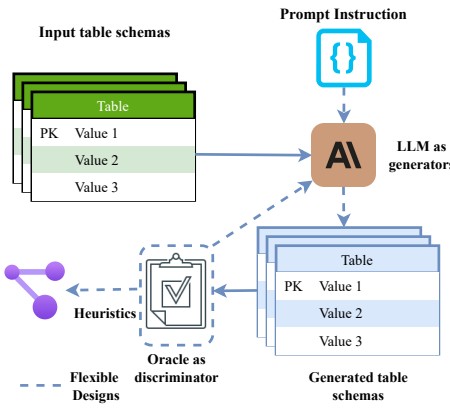

Figure 2: An illustration of our proposed AutoG framework.

tailed in Appendix D.1.1. The input context module compiles the following information for LLMs: 1. Input table schema with inferred column types, 2. Statistical summary of the data (e.g., sample count, uniqueness, mode, and randomly sampled values), 3.Task description for the downstream objective, 4. Chain-of-thought examples (further discussed in Section 4.2). The statistical summary

plays a pivotal role in guiding LLM decisions. Following Hollmann et al. (2023), we include: number of samples, number of unique samples, mode values, and randomly sampled column values. For example, a column where unique samples equal the total sample count is likely a primary key. LLMs generate concise column descriptions based on column names and these statistics. Additionally, we incorporate similarity scores from Dong et al. (2023) as prior knowledge, allowing LLMs to integrate statistics, textual descriptions, and similarity values holistically. As demonstrated in Section 5, this approach outperforms relying solely on DeepJoin's similarity scores (Dong et al., 2023).

**LLM as generators.** Based on input modules, we further leverage LLMs to generate a transformed schema. A straightforward approach is to let the LLM directly generate structured outputs such as YAML (Ben-Kiki et al., 2009)-formatted code. However, we find that open-ended generation usually produces invalid graph structures. To address this, inspired by the idea of function calling (Schick et al., 2024), we design basic augmentation actions based on 5 challenges of graph construction and then guide the output through chain-of-augmentation prompts, which is elaborated in Section 4.2.

**Oracle as discriminators.** After generating the graph schema, we adopt an oracle model to evaluate its effectiveness. First, we need a heuristic algorithm to convert tables into graphs. This paper considers two algorithms: Row2Node and Row2Node/Edge (Wang et al., 2024c). The former converts tables with at least two columns as FK and no PK into edges of a heterogeneous graph, while the latter converts other tables into nodes. We then train a GML model based on this graph and make final predictions. After generating the graph, we design an oracle as a discriminator to generate feedback. Such feedback can be qualitative, like whether graph construction fails due to grammar errors, or quantitative, like the performance of trained GML models. We discuss the oracle design in Section 4.3.

## 4.2 GUIDED GENERATION WITH CHAIN-OF-AUGMENTATION

The most straightforward way to let LLMs generate schema is directly generating the YAML-formatted structured outputs. However, such open-ended generation suffers from the following pitfalls: 1. LLMs generate schema and augmentation code with grammar errors, which makes the pipeline fail to proceed automatically. 2. LLMs tend to miss those node types and relations that require multi-step augmentation. Taking the `Diginetica` dataset as an example, relations may be found by first transforming set-attributed columns into proper augmented columns and then identifying the non `PK-FK` relations from the augmented columns. Simply generating the schema in a single-step manner fails to extract such relations. To alleviate these problems, we propose guided generation with a chain of augmentation. First, based on four challenges proposed in Section 3.1, we identify the following basic actions for augmentation.

1. `CONNECT_TWO_COLUMNS`: Building a `PK-FK` or `FK-FK` relationship between two columns. This action is designed to tackle challenge 1. Compared to joinable table discovery (JTD) (Dong et al., 2023; Hulsebos et al., 2019), LLMs can automatically identify the relationships without relying on humanly defined thresholds. Moreover, LLMs can also generate better join discovery results by utilizing meta information such as column statistics and descriptions. We validate this in Section 5.
2. `GENERATE_NEW_TABLE`: Inducing a new table from the original table via moving columns without changing any values. This can be viewed as identifying multiple node or relation types from the original table. This action is designed to tackle challenge 2. Moreover, this action can also be used to normalize the original data to satisfy certain normal forms (Ghosh et al., 2018), which enables our system to be self-contained without relying on external normalization tools.
3. `REMOVE(ADD)_PRIMARY_KEY`: Combined with proper heuristic methods, this action can change the type of table (as a node or an edge type) in the generated graph. This action is designed to tackle challenge 3.
4. `UNFOLD_MULTI_CATEGORY_COLUMNS`: This action is usually provided by data normalization tools. We include it to make our system self-contained. Its difference compared to traditional data normalization tools is that 1. LLMs should determine whether unfolding such columns is helpful; 2. LLMs need to determine the data types of the unfolded columns.

Then, LLMs will determine which actions to take by looking at input context introduced in Section 4.1 and *chain of thought demonstrations*. For each of these actions, we provide a chain-of-thought demonstration to showcase its usage. Specifically, we find that chain-of-thought (CoT)

prompts (Wei et al., 2022) are critical to action generation. As a motivating example, LLMs tend to merely find those columns with identical names to build non-`PK-FK` relationships without CoT. Only after introducing CoT demonstrations can LLMs utilize the statistics of columns to find more general non-`PK-FK` relationships with different column names. After generating the draft action based on provided information, we empirically find that adding a **self-reflection** step with prompt like "Please double check and fix any errors" can further improve LLMs' capability to generate proper actions. The complete prompt design can be found in Appendix D.1. To determine the termination step, we add a null action to the action space and set a hard threshold $T$ to limit the maximum number of actions, typically set to 10 for our proposed datasets.

## 4.3 DESIGNING ORACLE TO GENERATE FEEDBACK

After generating the schema candidates, we need an oracle to evaluate their effectiveness and thus choose the best schema. Despite LLM's capability to generate schemas based on prior knowledge, they cannot quantitatively predict how different schemas affect downstream task performance (Wang et al., 2024b). We thus need a graph-centric model to generate the feedback.

The main challenge of designing a quantitative oracle is to 1. efficiently obtain the performance estimation, 2. the performance measure should present a model-agnostic estimation of the graph schema's influence on downstream tasks. We first explore the possibility to **speedup the estimation process**: (1) *Condensating the graph* (Hashemi et al., 2024), improving the evaluation efficiency by training and testing on a smaller graph; (2) *Adopting an early-stage training metric*, such as the validation set performance. We

Table 2: Evaluating different oracles by quality and efficiency. For sampling, we set the ratio to 30%. For early-stage validation performance, we set to 10% of total epochs (should be set according to different datasets). The pre-processing is set as the basic unit; all other time is rounded to an integer.

| | Discrepancy | Training (node) | Training (link) | Process |
|---|---|---|---|---|
| **Full** | 0 | 29x | 300x | 1x |
| **Sampled** | 0.75 | 16x | 95x | 1x |
| **Actively sampled** | 0.75 | 16x | 95x | 3x |
| **Early metric** | 0.09 | 10x | 52x | 1x |

then compare these methods in terms of their effectiveness and efficiency. Specifically, we randomly sample three groups of schemas (in total 36, with distinguishable performance) from the proposed datasets. Then, we let different oracles generate orders for each group and measure the normalized Kendall's tau distance (Kumar & Vassilvitskii, 2010) to ones generated by regular GML models. From the experimental results in Table 2, we find that only the early-stage validation performance can reasonably estimate the downstream task performance. In AutoG, we use this strategy to speed up the estimation of some large-scale datasets. We then discuss developing a **model-agnostic performance estimation**. We find that sometimes there presents a gap between different GNN models such as RGCN (Schlichtkrull et al., 2018) and heterogeneous graph transformer (Hu et al., 2020b). To address this problem, we adopt a simple strategy to use the average performance adopted from a basket of GNN models as the final oracle score. In this paper, we consider RGCN (Schlichtkrull et al., 2018), RGAT (Veličković et al., 2018; Wang et al., 2024c), HGT (Hu et al., 2020b), and PNA (Corso et al., 2020) as the basket of GNN models, which are four widely adopted GNN models for heterogeneous graphs (Wang et al., 2024c).

## 4.4 GENERATING DIVERSE CANDIDATES

Despite AutoG's capability to generate graph schemas automatically, its decision relies on LLMs' reasoning and may not always generate optimal results (as shown in Section 5). One remedy for this issue is to let LLM generate a set of candidate results and then select the best based on performance measured by oracles. When AutoG is adopted as a tool for data scientists, diverse candidates can be post-processed to help them make decisions. To produce diverse schemas, we run the algorithm multiple times and choose the candidates with the best oracle score as the final selection. We denote such version of AutoG as **AutoG-A**, while the alternative directly uses the final state as **AutoG-S**. We demonstrate when **AutoG-S** is not good enough to produce the best graph schemas and what's the underlying reason in Section 5.

## 5 EXPERIMENTAL RESULTS

In this section, we systematically evaluate the AutoG framework on the proposed benchmarks from the following perspectives: 1. *Quantitative evaluation* of the performance of different graph construction methods. 2. *In-depth analysis* of the working mechanism of AutoG and its limitations.

### 5.1 EXPERIMENTAL SETTINGS

To investigate the impact of different graph construction methods, we fix the GML model to check the downstream task performance according to different graph schemas. Specifically, we commonly used baselines on heterogeneous graphs, including RGCN, RGAT, HGT, and RPNA (Wang et al., 2024c). This section presents RGCN's performance and shows others in Appendix E.1. We select Claude's Sonnet-3.5 as the backbone of LLMs and discuss the influence of different LLMs in Appendix E.2. We consider the following baseline methods: (1) XGBoost (Chen & Guestrin, 2016) and DeepFM (Guo et al., 2017), (2) TabGNN (Guo et al., 2021), (3) JTD with Row2Node/Edge (Dong et al., 2023; Gan et al., 2024), (4) Graph schema designed by human experts. We detail the expert schema design in Appendix E.3, (5) Vanilla graph schema: Based on the expert schema, we remove relations introduced by experts to construct a minimal relation subset supporting graph construction, and then generate relations with heuristics. It should be noted that JTD and TabGNN take an easier setting because they are augmented from the vanilla graph schema rather than the original schema without any information; otherwise, they can not even generate valid graphs.

### 5.2 QUANTITATIVE EVALUATION

Table 3 shows the performance of different graph construction methods. Models' performance is used to determine the quality of graphs. We consider both the individual model performance and the average model set's performance. We show the performance of RGCN, ranking of RGCN, and ranking of average performance in Table 3. The metrics for each task are shown in the second column, and the ranking is calculated based on the average ranking of each task. **To prevent shortcuts by looking at column names, we change all identical column names to different names.**

Table 3: Evaluation of different graph construction methods on proposed datasets. The best is in bold, second best is underlined, and third best is double-underlined.

| Dataset | Task | XGBOOST | DeepFM | TabGNN | Vanilla Schema | | JTD Schema | | AutoG | Expert |
|---|---|---|---|---|---|---|---|---|---|---|
| | | N/A | N/A | TabGNN | R2N | R2NE | R2N | R2NE | AutoG | Expert |
| **Datasets with a Single Downstream Task** | | | | | | | | | | |
| IEEE-CIS | Fraud (AUC) | 90.14 | **90.28** | 75.38 | 89.17 | 89.17 | 89.17 | 89.17 | 89.17 | 87.28 |
| RetailRocket | CVR (AUC) | 50.35 | 49.33 | 82.84 | 50.45 | 49.90 | 50.82 | 48.99 | 74.78 | **84.70** |
| Movielens | Ratings (AUC) | 53.62 | 50.93 | 55.34 | 57.34 | 61.20 | 54.55 | 57.23 | 69.10 | 69.73 |
| Outbrain | CTR (AUC) | 50.05 | 51.09 | 62.12 | 49.33 | 52.06 | 49.35 | 52.23 | 61.32 | 62.71 |
| AVS | Repeat (AUC) | 52.71 | 52.88 | 54.48 | 47.75 | 48.84 | 53.27 | 53.27 | **56.03** | 55.08 |
| **Datasets with Multiple Downstream Tasks** | | | | | | | | | | |
| | Venue (Acc) | 21.95 | 28.19 | 42.84 | 27.24 | 46.26 | 21.26 | 46.97 | **49.88** | 49.66 |
| MAG | Citation (MRR) | 3.29 | 45.06 | 70.65 | 65.29 | 65.29 | 72.53 | 81.50 | 80.84 | 80.86 |
| | Year (Acc) | 28.09 | 28.42 | 52.77 | **54.09** | 30.90 | 53.07 | 53.07 | **54.09** | 35.35 |
| Diginetica | CTR (AUC) | 53.50 | 50.57 | 50.00 | 68.44 | 65.92 | 50.05 | 50.00 | 75.07 | 75.07 |
| | Purchase (MRR) | 3.16 | 5.02 | 5.01 | 5.64 | 7.70 | 11.37 | 15.47 | 36.91 | 36.91 |
| Stackexchange | Churn (AUC) | 58.20 | 59.84 | 78.27 | 77.67 | 76.47 | 85.58 | 84.85 | 86.92 | 85.58 |
| | Upvote (AUC) | 86.69 | 87.64 | 85.28 | 86.45 | 86.47 | **88.61** | 67.98 | 87.43 | 88.61 |
| **Ranking (RGCN)** | | 5.9 | 5.2 | 4.7 | 4.2 | | 3.1 | | 2.0 | 2.0 |
| **Ranking (Average)** | | 5.8 | 5.3 | 4.5 | 4.4 | | 3.3 | | 2.0 | 2.0 |

From the experimental results, we make the following observations

- **AutoG generates high-quality graphs:** The AutoG method we propose can surpass other automatic graph construction methods and reach close to the level of human experts.
- **AutoG's superiority against heuristic-based methods:** Heuristic-based automatic discovery methods can only be applied to some special cases. We demonstrate its superiority against JTD with an example of MAG. When applying JTD to the MAG dataset. JTD ranks the column "paper_cite" from Table "Cites" and column "paper_writer" from Table "Writes" as the second most

similar pairs, which is incorrect. AutoG avoids this problem by making decisions based on meta-data provided in context.

• **The same graph may not be adequate for different downstream tasks.** On the MAG dataset, we observe that the expert-designed graph is not optimal for the year prediction task and is much worse than the original schema. We discuss this problem in more detail in Section 5.3.1.

### 5.3 IN-DEPTH ANALYSIS

To better understand the effectiveness of AutoG, we further study the effect of its components. We conduct three experiments: (1) Comparing AutoG variants, and studying when AutoG or expert schema fails to deliver promising downstream task performance. (2) Studying the necessity of each prompt component and AutoG's performance on synthetic data with anonymous columns.

### 5.3.1 STUDYING AUTOG VARIANTS

We consider variants of AutoG: AutoG-S and AutoG-A, where AutoG-S directly adopts the final output state while AutoG-A uses an oracle to select the state. We also consider AutoG-O, which conducts open-ended generation. As shown in Table 4, we draw the following conclusions: 1. Close-ended generation is necessary for valid schema generation. 2. Oracle is often unnecessary, meaning LLMs can gener-

Table 4: Ablation studies for closed-ended generation and oracles

| Dataset | Task | Valid | | | Performance | |
|---|---|---|---|---|---|---|
| | | AutoG-O | AutoG-S | Auto-A | AutoG-S | Auto-A |
| MAG | Venue | ✗ | ✓ | ✓ | 49.88 | 49.88 |
| | Year | ✗ | ✓ | ✓ | 50.99 | 54.09 |
| IEEE-CIS | Fraud | ✗ | ✓ | ✓ | 87.28 | 89.17 |
| RetailRockets | CVR | ✗ | ✓ | ✓ | 74.78 | 74.78 |

ate good candidates merely based on prior knowledge. However, two bad cases exist: (1) **Expert schema doesn't work well**: An example is the year prediction task on the MAG dataset. Taking a deeper look at the generated graph statistics, we find that when predicting the venue of "Paper", the adjusted homophily (Lim et al., 2021) of labels based on metapath "Paper-Author-Paper" is 0.156. While for year prediction, the adjusted homophily is only 0.02. This can be viewed as an extension of the heterophily problem (Lim et al., 2021) towards the RDB data, and an effective graph construction algorithm should address this problem by eliminating harmful relations. Although we deliberately introduce chain-of-thought prompts in the graph construction, AutoG still needs to rely on a graph oracle to deal with this problem. (2) **All graph construction methods don't work well**: An example is the IEEE-CIS dataset. We find that despite the reasonable data normalization process (for example, generating a new table based on columns forming an independent entity "MatchStatus", full example can be shown in Appendix E.3), such graph construction negatively affects the performance of GML model compared to the original one. This phenomenon corresponds to a more challenging scenario in which we must infer beneficial network effect (Lee et al., 2024). Compared to the homophily/heterophily problem, which may be mitigated by calculating graph statistics, inferring network effects is much more challenging, especially when the semantics of columns can't tell network effects. Generally, determining whether a graph is beneficial and telling whether a graph can be constructed are key challenges when applying GML to industrial data.

### 5.3.2 WORKING MECHANISM OF AUTOG

Despite the promising performance of AutoG, *LLM as generators* is composed of complicated prompt designs, which makes it challenging to understand the role of each component and how they may be applied to more general types of tabular data (for example, ones with anonymous columns). We thus further study the influence of different prompt components. In our prompt design, we have considered the following components: 1. the semantic information of the

Table 5: Ablation studies of different AutoG prompt components. "Orig" stands for the original schema with original names. "Anon" stands for the anonymous column names. "3/3" means 3 of the 3 expected actions have all been generated.

| | Challenge 1 | | Challenge 2 | | Challenge 3 | |
|---|---|---|---|---|---|---|
| | Orig | Anon | Orig | Anon | Orig | Anon |
| **Default** | 3/3 | 1/3 | 2/3 | 1/3 | 2/2 | 0/2 |
| **No COT** | 1/3 | 0/3 | 1/3 | 0/3 | 0/2 | 0/2 |
| **No stats** | 1/3 | 0/3 | 1/3 | 0/3 | 0/2 | 1/2 |
| **No demo** | 0/3 | 0/3 | 0/3 | 0/3 | 0/2 | 0/2 |

column (column name); 2. the statistical meta-information of the column; 3. the examples given in the prompt; 4. the chain of thought demonstrations for each action. Specifically, we built a synthetic dataset based on MAG to include the challenges $1 - 4$ proposed in Section 3.1 and ensure the test data is not included in the pre-training set of LLMs. Compared to quantitative evaluation, we di-

rectly study whether LLMs can generate the required actions for better graphs. As shown in Table 5, we observe the following conclusions: 1. Demonstration is necessary for AutoG to generate valid actions. 2. Both COT and statistics are critical to the graph schema generation. Specifically, we find that LLMs will only find trivial augmentations (for example, non-`PK-FK` relations with identical column names), which means COT is the key for LLMs to conduct deep reasoning and to utilize the statistics sufficiently. 3. Semantic information of the column names is vital for the performance of AutoG, which is a limitation of AutoG.

## 6 RELATED WORKS

Recently, GML has been widely adopted to capture the structural relationship across tabular data (Li et al., 2024). One of the key challenges lies in identifying graph structures from tabular data that can benefit the downstream tasks. Early endeavors in database management mine relationships across databases using rule-based methods (Yao & Hamilton, 2008; Liu et al., 2012; Abedjan et al., 2015; Koutras et al., 2021). One limitation of these methods lies in their scalability towards large-scale tables. The rise of machine learning has led to two ML-based approaches: heuristic-based and learning-based methods. *Heuristic-based methods* transform tabular data into graphs based on specific rules. For instance, Guo et al. (2021) generates edge relationships based on columns with categorical values in the table, resulting in a multiplex graph through multiple columns. Wu et al. (2021) and You et al. (2020) create a bipartite graph based on each row representing a sample and each column representing a feature, where You et al. (2020) further supports numerical values by storing them as edge attributes. Du et al. (2022) generates a hypergraph by treating each row as a hyperedge. A major challenge for these heuristic methods is the inability to handle multi-table scenarios effectively. Row2Node (Fey et al., 2024) and Row2Node/Edge (Wang et al., 2024c) are proposed for multiple tables with explicit key relationships. Bai et al. (2021) designs an end-to-end model for RDB prediction tasks. These methods are still limited to tables satisfying RDB specifications. Learning-based methods aim to learn edge relationships automatically based on the correlation between features. Chen et al. (2020) and Franceschi et al. (2019) leverage graph structure learning to learn the induced edge relationships between each sample. However, learning-based methods suffer from efficiency issues, and their effectiveness is challenged by Errica (2024) when adequate supervision is provided. Koutras et al. (2020) leverages knowledge graph to build relation graph across different columns and extract potential structural relationships. Dong et al. (2023) leverages a language model embedding to detect similar columns in the table and thus extract those related columns. *To study the effectiveness of different GML methods for tabular data*, multiple benchmarks have been developed (Wang et al., 2024c; Fey et al., 2024; Bazhenov et al., 2024). However, their scopes are limited to either model evaluation (Wang et al., 2024c; Fey et al., 2024) or feature evaluation (Bazhenov et al., 2024), which leaves graph construction evaluation an underexplored area.

## 7 CONCLUSION

In this work, we formalize the graph construction problem through a benchmark and propose AutoG, an LLM-based solution for automated graph generation. Our extensive experiments demonstrate that graph construction critically impacts downstream task performance. However, automatic graph construction remains highly challenging; AutoG serves as a preliminary step, currently addressing only moderately complex scenarios. Looking ahead, we identify three fundamental challenges pivotal to this field: (1) establishing criteria to evaluate whether a graph structure offers measurable advantages over non-graph methods; (2) determining the feasibility of deriving a beneficial graph from multi-tabular datasets; and (3) isolating core relational patterns essential for task performance while pruning superfluous connections. Resolving these challenges is imperative to bridge the gap between theoretical GML advancements and their robust, scalable application in real-world industrial settings.

## 8 ACKNOWLEDGMENTS

We thank the authors of 4DBInfer (Wang et al., 2024c) for open-sourcing their project, which has been instrumental in enabling us to develop the codebase for graph construction.

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

# A  MORE PRELIMINARIES

## A.1  DATA TYPES

In this paper, we consider the following data types {`category`, `numeric`, `text`, `primary_key`(PK), `foreign_key` (FK), `set`, `timestamp`}.

- `category`: A data type representing categorical values. For example, a column with three possible values ("Book", "Pen", "Paper") is of the `category` data type.
- `numeric`: A data type representing numerical values. This can include integers, floating-point numbers, or decimals. For instance, a column storing ages or prices would typically be of the `numeric` data type.
- `text`: A data type representing textual data. This can include strings of characters, sentences, or even paragraphs. A column storing product descriptions or customer reviews would be of the `text` data type.
- `primary_key` (PK): A special type of column or a combination of columns that uniquely identifies each row in a table. It ensures data integrity and is often used to establish relationships between tables.
- `foreign_key` (FK): A column or a combination of columns in one table that refers to the `primary_key` in another table. It creates a link between the two tables, enabling data relationships and maintaining consistency.
- `set`: A data type representing a collection of values. It is often used to store multiple choices or options associated with a particular record.

- `timestamp`: A data type representing time. It's used to define the time-based neighbor sampler and prevents data leakage.

## A.2 EXAMPLES OF DATA FORMATS

We follow Wang et al. (2024c) to represent the table schema as a YAML-formatted configuration file. An example is shown below. An example original schema plot is shown in Figure 3. The original schema only presents limited relations, which may result in an ineffective graph for downstream tasks. Figure 4 shows an example of augmented relations schemas. With augmented tables including Company, Brand, Category, Customer, and Chain, the resulting graphs will benefit downstream tasks.

```
 1  tables:
 2    - name: History
 3      source: data/history.pqt
 4      format: parquet
 5      columns:
 6        - name: chain
 7          dtype: category
 8        - name: market
 9          dtype: category
10        - name: offerdate
11          dtype: datetime
12        - name: id
13          dtype: primary_key
14        - name: repeater
15          dtype: category
16        - name: offer
17          dtype: foreign_key
18          link_to: Offer.offer
19      time_column: offerdate
20  ......
```

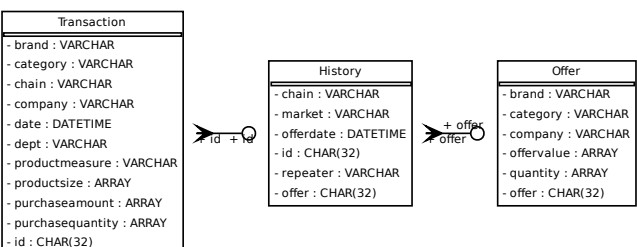

Figure 3: The original schema for the dataset AVS

## B DATASETS

`Movielens` is a collection of movie ratings and tag applications from MovieLens users. This dataset is widely used for collaborative filtering and recommender system development. We adopt the tabular version from the original website. Expert schema is designed by ourselves.

`MAG` is a heterogeneous graph dataset containing information about authors, papers, institutions, and fields of study. We adopt the tabular version from Wang et al. (2024c) and generate the original version by removing relations added by experts.

`AVS` (Acquire Valued Shoppers) is a Kaggle dataset predicting whether a user will repurchase a product based on history sessions.

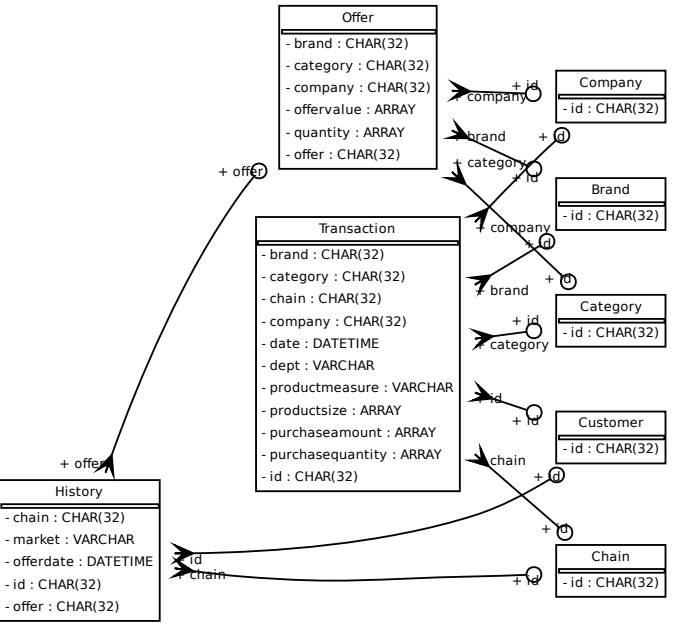

Figure 4: The new schema for dataset AVS with augmented relations

`IEEE-CIS` is a Kaggle dataset predicting whether a transaction is fraudulent. We adopt the original version from the website. Expert schema is designed by ourselves.

`Outbrain` is a Kaggle dataset predicting which pieces of content its global base of users are likely to click on.

`Diginetica` is a Codalab dataset for recommendation system.

`Retailrocket` is a Kaggle dataset for recommender system.

`Stackexchange` is a database from Stackexchange platform.

## C   MORE RELATED WORKS

**LLMs for automated data science.** Our work is also related to applying LLMs to automated data science. The core principle of these works lies in adopting the code generation capabilities of LLMs to automatically generate code for data curation (Chen et al., 2023), data augmentation (Hollmann et al., 2023), or working as a general interface for diverse data manipulation (Zhang et al., 2023; Hong et al., 2024; Hassan et al., 2023). Zhang et al. (2024) proposes a benchmark to evaluate the capabilities of LLMs in various data science scenarios. Compared to the methods adopted in these works, AutoG adopts close-ended generation via function calling to ensure the correctness of generation. Besides black-box LLMs, Suhara et al. (2022) fine-tunes and utilizes pre-trained language models on various data profiling tasks, such as column annotation.

**Learning on heterogeneous graphs** Heterogeneous graphs featuring multiple node and edge types naturally abstract relational database data. Learning representations within these graphs often rely on meta-paths Yang et al. (2020), which transform heterogeneous relations into homogeneous sets. Early methods focused on similarity measures derived from meta-paths Sun et al. (2011). With the advent of Graph Neural Networks (GNNs), approaches like HAN (Wang et al., 2019) extract multiple homogeneous graphs based on meta-paths for individual encoding. MAGNN (Fu

et al., 2020) further accounts for the roles of intermediate nodes in meta-paths. Alternatively, RGCN (Schlichtkrull et al., 2018) and G2S (Beck et al., 2018) emphasize relational graphs, where edges carry rich semantic information.

# D MORE DETAILS ON METHODS

## D.1 PROMPT DESIGN

Our prompt design is demonstrated below. The key prompts we use include: 1. prompt used to infer the initial type of table columns 2. prompt used to generate augmentation actions

### D.1.1 DATA TYPE INFERENCE PROMPT

```
 1     Now you will be given a list of tables and columns, each one with the
           following format:
 2 Analysis for Table <name of the table>:
 3   Column: <name of the column 1>
 4     Max: <max value of the column>
 5     Min: <min value of the column>
 6     Mode: <mode value of the column>
 7     Sampled Values: <list of sampled values>, for example, ['value1', '
       value2', 'value3']
 8   Column: <name of the column 2>
 9     Max: <max value of the column>
10     Min: <min value of the column>
11     Mode: <mode value of the column>
12     Sampled Values: <list of sampled values>, for example, ['value1', '
       value2', 'value3']
13 ...
14
15 You should identify the data type of each column. The data types you can
       choose from are:
16 ['float', 'category', 'datetime', 'text', 'multi\_category']
17 float: The column is probably a float-type embedding tensor. There should
        be (nearly) no redundant values.
18 category: The column is probably a categorical column.
19 datetime: The column is probably a datetime column. Only full datetime
       values should be considered, some columns presenting only year or
       month or day should be better considerd as category.
20 text: The column is probably a text column. There should be a lot of
       unique values. Otherwise it will probably be a category column.
       Moreover, we should expect texts with natural semantics, otherwise it
       's probably a category column.
21 multi_category: The column is probably a multi-category column. Usually
       this means the column value is a list.
22 It should be noted that if the column is probably an embedding type, then
        directly put it to the float type.
23 Then, you should output a discription of the column, for example:
24 "This column is probably representing the ID from 1 to n of users in the
       system, as it has a lot of unique values."
25 Output the results with the following format:
26 {
27     "<name of the table>": {
28         "<name of the column 1>": ("<data type of the column 1>", "<
       description of the column 1>"),
29         "<name of the column 2>": ("<data type of the column 2>", "<
       description of the column 2>")
30     },
31     ...
32 }
33
34 In description, if you see two columns are very similar and may represent
        the same thing, you should mention it.
```

### D.1.2 Augmentation generation prompt

```
1  Imagine you are an expert graph data scientist, and now you are expected
       to construct graph schema based on the original
2  inputs. You will be given an original schema represented in the
       dictionary format:
3  <data>
4      1. dataset\_name: name of the dataset
5      2. tables: meta data for list of tables, each one will present
       following attributes
6          1. name: table name
7          2. source: source of the data, can either be a numpy .npz file or
        a parquet file
8          3. columns: list of columns, each column will have following
       attributes
9              1. name: column name
10             2. dtype: column type, can be either text, categorical, float
       , primary\_key, foreign\_key, or multi\_category. primary\_key and
       foreign_key are two special types of categorical columns, which
       presents a structural relationship with other tables. Multi\_category
        means this column is of list type, and each cell main contains a
       list of categorical values. After a column is set as primary\_key or
       foreign\_key, it should not be changed to other types.
11             3. link\_to (optional): if this column is a foreign key,
       point to which primary key from which table
12      3. statistics of the table: statistics of the column value of tables.
        These statistics can be used to help you
13      determine the characteristics of the columns. For example, if one
       categorical column only contains one unique value,
14      then creating a node type based on this column can result in a super
       node, which is not ideal for graph construction.
15      You should also determine whether two columns represent the same
       thing based on these statistics.
16      4. Dummy table is a special type of table. It's not explicitly
       defined with a table slot. It's defined in other tables, such as
17      {{"name": "nation", "dtype": "foreign\_key", "link\_to": "Country.
       CountryID"}}. In this case, "Country" is a dummy table, which is not
18       explicitly defined in the tables slot.
19  </data>
20  Here are the documents of the actions:
21
22  {actions}
23
24
25  Now, you need to
26  1. Actively think about whether any one of the four actions should be
       conducted; If not, you can select "None" and then halt the program.
27  2. output all actions you can think of from the above list to perform,
       and output your selection in the following format. It should be noted
        that for those actions with sequential relation like one new
       categorical column generated after expanding a multi-category column,
        you don't need to generate in one round.
28
29  <selection>
30  [{{'explanation': <explanation for the selection>, 'action': <first
       action>, 'parameters': <parameters for the first action> }},
31  {{'explanation': <explanation for the selection>, 'action': <second
       action>, 'parameters': <parameters for the second action> }}, ...
32  ]
33  </selection>
34
35
36  3. If not more action, output <selection>None</selection>
37
38  Example:
```

```
39 {example}
40
41 History Actions:
42 {history\_actions}
43
44 <input>
45 <dataset\_stats>
46 {stats}
47 </dataset\_stats>
48 <task>
49 {task}
50 </task>
51 <schema>
52 {input\_schema}
53 </schema>
54 Here we gives the similarity score of each column pair, you can use this
       information to determine whether two columns may be joinable. The
       similarity score is scaled to [0, 1], the larger means the more
       similar.
55 <similarity>
56 {jtd}
57 </similarity>
58 </input>
59 Return your output in the json format inside <selection></selection>.
```

Specifically, there are five key components for an action generation prompt.

First, we give the document of each implemented action.

```
 1 Here is the introduction of the function generate_or_connect_dummy_table:
 2 Description:
 3 This function can be used in two ways:
 4 1. Generate a dummy table with only one primary key
 5 2. Turn an existing column with categorical type to an existing dummy
       table
 6 "orig_col_name" must be a column with category type
 7 Parameters:
 8 dbb: the database object
 9 base_table_name: the name of the original table
10 orig_col_name: the name of the original column in the original table,
      this should be a column with category type
11 new_table_name: the name of the new table to be created/connected
12 new_col_name: the name of the new column to be created/connected
13
14 Here is the introduction of the function connect_two_columns:
15 Description:
16 Connect two columns, this function can be used for the following case.
       Always put the column with category type in table 1.
17 1. A category column in table 1 is connected to a category column in
       table 2, in this case, a new dummy table will be created
18 2. A category column in table 1 is connected to a primary key column in
       table 2, in this case, the column in table 1 will be turned into a
       foreign key column. In case 2, table_2_col_name must be a primary key
        column
19 3. A category column in table 1 is connected to a non-category and non-
       primary key column in table 2, in this case, we will use a trick
       called Surrogate Key.
20 4. If the column in table 1 is already a foreign key, then in this case
       it's probably a multi-column-point-to-one case, we need to update
       other fk columns too.
21 Parameters:
22 dbb: the database object
23 table_1_name: the name of the first table,
24 table_1_col_name: the name of the column in the first table, this should
      be a column with category type
```

```
25 table_2_name: the name of the second table
26 table_2_col_name: the name of the column in the second table, this should
       be a column with category type
27
28 Here is the introduction of the function explode_multi_category_column:
29 Description:
30 Explode a multi-category column into multiple columns. You should
       determine whether to use this function. If you don't explode a multi-
       category column, it will be treated as a single category column
       automatically.
31 Parameters:
32 dbb: the database object
33 original_table: name of the original table where the multi-category
       column is located
34 multi_cat_col: the name of the multi-category column
35 primary_key_column: the name of the primary key column in the original
       table
36 new_table_name: the name of the new table to be created
37 new_col_name: the name of the new column to be created
38 dtype: the data type of the new column, if set to "foreign_key", this
       table will contain only "foreign_keys". In this case, it means you
       only want to use this column's relaion. If set to other types, this
       table will contain the original column's values, and a primary key
       will be added, this means you want to use this column's values.
39
40 Here is the introduction of the function generate_non_dummy_table:
41 Description:
42 Generate a non-dummy table with columns in the original table
43 Parameters:
44 dbb: the database object
45 base_table_name: the name of the original table
46 cols: the list of columns to be included in the new table and removed
       from the original table
47 new_table_name: the name of the new table to be created
48
49 Here is the introduction of the function remove_primary_key:
50 Description:
51 Remove a primary key constraint from a column in the original table
52 If the column is just an index, then the column will be removed from the
       table.
53 For example, if the schema is like {
54     {"name": "id", "dtype": "primary_key"},
55     {"name": "user", "dtype": "foreign_key", "link_to": "user.userID"},
56     {"name": "book", "dtype": "foreign_key", "link_to": "book.bookID"},
57 }
58 In such case, it's clear that this table represents the role of an edge,
       while the presence of primary key prevents heuristic to turn this
       table into an edge. Primary key is not needed in this case.
59 In such case, we will remove the primary key constraint from the column.
60 Parameters:
61 dbb: the database object
62 base_table_name: the name of the original table
63 col_name: the name of the column in the original table
64
65 Here is the introduction of the function add_primary_key:
66 Description:
67 Add a primary key column to the original table
68 Parameters:
69 dbb: the database object
70 base_table_name: the name of the original table
71 col_name: the name of the newly added primary key column
```

Second, for in-context learning examples, we use the following prompt.

```
1 Table: Paper
```

```
 2  {
 3    "Column": "PaperID",
 4    "data type": "primary_key"
 5  }
 6  {
 7      "Column": "Title",
 8      "data type": "text",
 9      "Number of unique values": 10000,
10      "Number of nan values": 0,
11      "Number of total values": 10000,
12      "Mode values": "Transformers",
13      "5 sampled values": [
14          "Transformers",
15          "Graph Neural Networks",
16          "Reinforcement Learning",
17          "Meta Learning",
18          "Computer Vision"
19      ]
20  }
21  {
22      "Column": "Authors",
23      "data type": "multi_category",
24      "Number of unique values": 987,
25      "Number of nan values": 0,
26      "Number of total values": 74320,
27      "Mode values": "Yann LeCun",
28      "5 sampled values": [
29          "Yann LeCun",
30          "Geoffrey Hinton",
31          "Yoshua Bengio",
32          "Fei-Fei Li",
33          "Jitendra Malik"
34      ]
35  }
36  {
37      "Column": "Journal",
38      "data type": "category",
39      "Number of unique values": 100,
40      "Number of nan values": 0,
41      "Number of total values": 10000,
42      "Mode values": "Nature",
43      "5 sampled values": [
44          "Nature",
45          "Science",
46          "NeurIPS",
47          "ICML",
48          "CVPR"
49      ]
50  }
51  {
52      "Column": "Year",
53      "data type": "float",
54  }
55  {
56      "Column": "Keywords",
57      "data type": "category",
58      "Number of unique values": 100,
59      "Number of nan values": 0,
60      "Number of total values": 10000,
61      "Mode values": "Machine Learning",
62      "5 sampled values": [
63          "Machine Learning",
64          "Deep Learning",
65          "Graph Neural Networks",
66          "Reinforcement Learning",
```

```
 67          "Meta Learning"
 68      ]
 69 }
 70 {
 71      "Column": "Abstract",
 72      "data type": "text",
 73      "Number of unique values": 10000,
 74      "Number of nan values": 0,
 75      "Number of total values": 10000,
 76      "Mode values": "This paper presents a new model for graph neural
            networks.",
 77      "5 sampled values": [
 78          "This paper presents a new model for graph neural networks.",
 79          "This paper introduces a new reinforcement learning algorithm.",
 80          "This paper presents a new model for transformers.",
 81          "This paper presents a new model for meta learning.",
 82          "This paper presents a new model for computer vision."
 83      ]
 84 }
 85 {
 86      "Column": "Category",
 87      "data type": "category",
 88      "Number of unique values": 10,
 89      "Number of nan values": 0,
 90      "Number of total values": 10000,
 91      "Mode values": 3,
 92      "5 sampled values": [
 93          3,
 94          4,
 95          1,
 96          6,
 97          9
 98      ]
 99 }
100 {
101    "Column": "ItemID",
102    "data type": "foreign_key"
103 }
104 Table: Journal
105 {
106    "Column": "JournalID",
107    "data type": "primary_key"
108 }
109 {
110    "Column": "Name",
111    "data type": "text",
112      "Number of unique values": 100,
113      "Number of nan values": 0,
114      "Number of total values": 100,
115      "Mode values": "Nature",
116      "5 sampled values": [
117          "Nature",
118          "Science",
119          "NeurIPS",
120          "ICML",
121          "CVPR"
122      ]
123 }
124 {
125      "Column": "ImpactFactor",
126      "data type": "float"
127 }
128 {
129      "Column": "Country",
130      "data type": "category",
```

```
131        "Number of unique values": 10,
132        "Number of nan values": 0,
133        "Number of total values": 100,
134        "Mode values": "USA",
135        "5 sampled values": [
136            "USA",
137            "USA",
138            "Canada",
139            "UK",
140            "USA"
141        ]
142 }
143 {
144        "Column": "Publisher",
145        "data type": "text",
146        "Number of unique values": 9,
147        "Number of nan values": 0,
148        "Number of total values": 100,
149        "Mode values": "Springer",
150        "5 sampled values": [
151            "Springer",
152            "Elsevier",
153            "ACM",
154            "IEEE",
155            "Nature"
156        ]
157 }
158 {
159        "Column": "PublisherLocation",
160        "data type": "category",
161        "Number of unique values": 5,
162        "Number of nan values": 0,
163        "Number of total values": 100,
164        "Mode values": "USA",
165        "5 sampled values": [
166            "USA",
167            "USA",
168            "Canada",
169            "UK",
170            "USA"
171        ]
172 }
173
174 </dataset_stats>
175 <tasks>
176 Now I want to train a model which can predict the category of a paper
        based on the information in the paper.
177 </tasks>
178 <schema>
179 {
180        "dataset_name": "Papers",
181        "tables": [
182            {
183                "name": "Paper",
184                "source": "data/paper.npz",
185                "columns": [
186                    {"name": "PaperID", "dtype": "primary_key"},
187                    {"name": "Title", "dtype": "text"},
188                    {"name": "Authors", "dtype": "multi_category"},
189                    {"name": "Journal", "dtype": "category"},
190                    {"name": "Year", "dtype": "float"},
191                    {"name": "Keywords", "dtype": "category"},
192                    {"name": "Abstract", "dtype": "text"},
193                    {"name": "Category", "dtype": "category"}
194                ]
```

```
195                },
196                {
197                    "name": "Journal",
198                    "source": "data/journal.npz",
199                    "columns": [
200                        {"name": "JournalID", "dtype": "primary_key"},
201                        {"name": "Name", "dtype": "text"},
202                        {"name": "ImpactFactor", "dtype": "float"},
203                        {"name": "Country", "dtype": "category"},
204                        {"name": "Publisher", "dtype": "text"},
205                        {"name": "PublisherLocation", "dtype": "category"}
206                    ]
207                }
208            ]
209        }
210 </schema>
211 Here we gives the similarity score of each column pair, you can use this
        information to determine whether two columns may be joinable. The
        similarity score is scaled to [0, 1], the larger means the more
        similar.
212 <similarity>
213 The pair with the 1st highest similarity is column "Journal" from Table "
        Paper" and column "Name" from Table "Journal" with similarity 0.885
214 The pair with the 2nd highest similarity is column "Authors" from Table "
        Paper" and column "Name" from Table "Journal" with similarity 0.743
215 The pair with the 3rd highest similarity is column "Authors" from Table "
        Paper" and column "Country" from Table "Journal" with similarity
        0.723
216 </similarity>
217 </input>
218
219
220
221 We need to think about whether we need to do one of the six actions:
222 1. First, for explode_multi_category_column, the Authors of the paper are
         in a multi-category column. Moreover, author is closely related to
        the category of the paper, so the relationship Paper-Author-Paper can
         be very useful. So, we need to explode this multi category column.
223 2. For connect_two_columns, the Journal column in the Paper table and the
         column Name in the Journal table are highly similar, so we can
        connect these two columns with a foreign key constraint. Other pairs
        like Authors and Name, Authors and Country are not similar enough to
        be connected.
224 3. For generate_non_dummy_table, the Publisher and PublisherLocation
        columns are independent columns for the entity Publisher. We can
        generate a new table Publisher with these two columns.
225 4. For generate_or_connect_dummy_table, we need to find those categorical
         columns beneficial for downstream task. We have categorical columns
        (Journal has been deleted in step 2, Category is the final objective)
         Keyword, Country, this will result in relationship Paper-Keyword-
        Paper and Paper-Journal-Country-Journal-Paper respectively. Since the
         target is to predict the category of a paper, we can generate a
        dummy table for the column Keyword since paper sharing the same
        keyword are highly likely to share the same category. Country may be
        not beneficial since it doesn't present a strong semantic
        relationship with the category.
226 5. For remove_primary_key and add_primary_key, there's no unreasonable
        primary key or missing primary key in the table, so we don't need to
        do this action. as a result, we have the following actions
227 <selection>
228        [{{'explanation': "Author is multi-category and Paper-Author-
        Paper is probably useful. We set the dtype to foreign_key because we
        want to use the relation", 'action': 'explode_multi_category_column',
         'parameters': {'original_table': 'Paper', 'multi_cat_col': 'Author',
```

```
        primary_key_column: 'PaperID', 'new_table_name': 'Author', '
        new_col_name': 'AuthorName', 'dtype': 'foreign_key'}},
229         {{'explanation': 'the Journal column in the Paper table and the
        column Name in the Journal table are highly similar, both of them
        should refer to the name of the journal', 'action': '
        connect_two_columns', 'parameters': {'table_1_name': 'Paper', '
        table_1_col_name': 'Journal', 'table_2_name': 'Journal', '
        table_2_col_name': 'Name', 'new_table_name': "", 'new_table_col_name
        ': "" }},
230         {{'explanation': 'Publisher and PublisherLocation are independent
         columns for the entity Publisher. We can generate a new table
        Publisher with these two columns', 'action': '
        generate_non_dummy_table', 'parameters': {'base_table_name': 'Paper',
         'cols': ['Publisher', 'PublisherLocation'],  'new_table_name': '
        Publisher'}},
231         {{'explanation': 'Keyword is a categorical column which can be
        used to generate a dummy table. Country is not beneficial for the
        downstream task', 'action': 'generate_or_connect_dummy_table', '
        parameters': {'base_table_name': 'Paper', 'orig_col_name': 'Keyword',
         'new_table_name': 'Keyword', 'new_col_name': 'Keyword'}},
232         ]
233     </selection>
```

The third component reflects the statistics of each column's data. Specifically, following Dong et al. (2023), we consider the number of unique values, mode values, maximum and minimum values if numerical, and $k$ sampled values. If this column belongs to a list type, we will also consider how many unique values we will get after expanding it into a categorical column. We give an example as follows.

```
1  Analysis for Table paper:
2    Column: paperID
3      Max: 736388
4      Min: 0
5      Mode: 0
6      Sampled Values: [311458 138871 636067 119201 468996]
7      Number of Unique Values: 736389
8    Column: label
9      Max: 348
10     Min: 0
11     Mode: 1
12     Sampled Values: [190  85  45 183 283]
13     Number of Unique Values: 349
14   Column: feat
15 Column is multi-dimensional. Probably an embedding type. Usually not of
      interest
16   Column: year
17     Max: 2019
18     Min: 2010
19     Mode: 2013
20     Sampled Values: [2014 2010 2019 2010 2010]
21     Number of Unique Values: 10
22
23 Analysis for Table Cites:
24   Column: paper_cite
25     Max: 736388
26     Min: 0
27     Mode: 732008
28     Sampled Values: [571834 223729 711055  26073 352954]
29     Number of Unique Values: 617924
30   Column: paper_cited
31     Max: 736388
32     Min: 0
33     Mode: 428523
34     Sampled Values: [557047 417778 395162 521944 108483]
35     Number of Unique Values: 629169
```

```
36
37 Analysis for Table HasTopic:
38   Column: paper_name
39     Max: 736388
40     Min: 0
41     Mode: 69985
42     Sampled Values: [388977 701406 503766 451820 101399]
43     Number of Unique Values: 736389
44   Column: field_of_study
45     Max: 59964
46     Min: 0
47     Mode: 14055
48     Sampled Values: [12834 15397 21310 24376  3744]
49     Number of Unique Values: 59965
50
51 Analysis for Table AffiliatedWith:
52   Column: author
53     Max: 1134648
54     Min: 0
55     Mode: 244427
56     Sampled Values: [377682 380472 116413 434611 284604]
57     Number of Unique Values: 852987
58   Column: institution
59     Max: 8739
60     Min: 0
61     Mode: 649
62     Sampled Values: [2315  649 4029 5285  664]
63     Number of Unique Values: 8740
64
65 Analysis for Table Writes:
66   Column: paper_writer
67     Max: 1134648
68     Min: 0
69     Mode: 239580
70     Sampled Values: [613331 153535 540376 618466 462598]
71     Number of Unique Values: 1134649
72   Column: arxiv_id
73     Max: 736388
74     Min: 0
75     Mode: 522277
76     Sampled Values: [731086 691749 540097 711055 194402]
77     Number of Unique Values: 736389
```

The next component refers to the specific task instruction given by users. As an example, the instruction for the "citation" task on the "Mag" dataset is "This task is to predict the venue of a paper given the paper's title, abstract, authors, and publication year. You may use the meta relations between papers, authors, topics, and institutions to improve the performance.

Finally, we use the result from deepjoin as a prior in the generation prompt. It will be computed based on the pairwise cosine similarity. Specifically, an example is given as follows

```
1 The pair with the 1st highest similarity is column "PostId" from Table "
    Comments" and column "PostId" from Table "PostLink" with similarity
    0.964
2 The pair with the 2nd highest similarity is column "PostId" from Table "
    PostHistory" and column "PostId" from Table "PostLink" with
    similarity 0.950
3 The pair with the 3rd highest similarity is column "PostId" from Table "
    Comments" and column "PostId" from Table "PostHistory" with
    similarity 0.937
4 The pair with the 4th highest similarity is column "PostId" from Table "
    PostHistory" and column "PostId" from Table "Vote" with similarity
    0.928
5 The pair with the 5th highest similarity is column "PostId" from Table "
    PostLink" and column "PostId" from Table "Vote" with similarity 0.917
```

```
6 The pair with the 6th highest similarity is column "PostId" from Table "
      Comments" and column "PostId" from Table "Vote" with similarity 0.897
7 The pair with the 7th highest similarity is column "ExcerptPostId" from
      Table "Tag" and column "WikiPostId" from Table "Tag" with similarity
      0.890
8 ...
```

Moreover, we would like to claim that the information provided by Deepjoin is noisy. For example, on the MAG dataset

```
1 The pair with the 1st highest similarity is column "paper_cite" from
      Table "Cites" and column "paper_cited" from Table "Cites" with
      similarity 0.885
2 The pair with the 2nd highest similarity is column "paper_cite" from
      Table "Cites" and column "paper_writer" from Table "Writes" with
      similarity 0.840
3 The pair with the 3rd highest similarity is column "paper_cited" from
      Table "Cites" and column "paper_writer" from Table "Writes" with
      similarity 0.832
4 The pair with the 4th highest similarity is column "paper_cited" from
      Table "Cites" and column "paper_name" from Table "HasTopic" with
      similarity 0.822
5 The pair with the 5th highest similarity is column "paper_cite" from
      Table "Cites" and column "paper_name" from Table "HasTopic" with
      similarity 0.806
```

The "paper_writer" refers to the author of the paper, which is different from the "paper_cite" which refers to cited papers. However, LLMs can recover from such noise and generate robust results.

## E    MORE EXPERIMENTAL RESULTS

### E.1    RESULTS OF OTHER BASELINE MODELS

### E.2    INFLUENCE OF DIFFERENT LLMS

After experimenting with several different language models, we found that some models either completely fail to function, such as outputting augmentations of the examples provided in the prompt, or produce similar results. Specifically, we discovered that models like Sonnet 3.5 and those stronger than Sonnet 3.5, such as Sonnet 3 Opus, can serve as effective model backbones. In contrast, models like Llama 3.1 70B (Dubey et al., 2024) and Mistral 2 cannot produce valid results. If we infer based on ChatbotArena (Chiang et al., 2024) performance, we deduce that models stronger than Sonnet 3.5 can serve as effective backbones.

### E.3    DESIGN OF SCHEMAS

This section details the AutoG and expert schema design for each dataset we propose.

### E.3.1    IEEE-CIS

IEEE-CIS is a dataset with weak network effects. Specifically, we find that only two columns "billing region" and "billing country" can bring limited performance gain. However, AutoG-S and experts can't find such relations. AutoG-A outputs the original schema as the best output schema.

### E.3.2    RETAILROCKET

For Retailrocket dataset, AutoG doesn't find the relationship between Category and Item_Category table. In this case, it achieves sub-optimal performance compared to the expert ones.

Table 6: Experimental results on more backbone models.

| Dataset | Task | Backbone | TabGNN | Original schema | JTD schema | AutoG | Expert schema |
|---------|------|----------|--------|-----------------|------------|-------|---------------|
| | | | (Value) | (Value) | (Value) | (Value) | (Value) |
| **Datasets with single downstream task** | | | | | | | |
| IEEE-CIS | N/A | GAT | 74.65 | 87.23 | 87.23 | 87.23 | 87.43 |
| | | HGT | 75.82 | 89.97 | 89.97 | 89.97 | 87.42 |
| | | PNA | 75.49 | 89.14 | 89.14 | 89.14 | 86.66 |
| RetailRocket | N/A | GAT | 81.92 | 50.13 | 50.63 | 79.18 | 82.84 |
| | | HGT | 83.25 | 49.06 | 49.92 | 71.37 | 84.95 |
| | | PNA | 82.99 | 50.43 | 50.95 | 82.59 | 84.27 |
| Movielens | N/A | GAT | 54.78 | 56.42 | 62.98 | 74.81 | 75.96 |
| | | HGT | 64.12 | 60.12 | 63.46 | 74.42 | 74.63 |
| | | PNA | 63.23 | 62.66 | 62.20 | 74.33 | 74.75 |
| Outbrain | N/A | GAT | 62.44 | 52.54 | 52.73 | 61.57 | 63.08 |
| | | HGT | 62.58 | 52.13 | 52.78 | 61.83 | 63.22 |
| | | PNA | 62.63 | 52.74 | 52.98 | 61.75 | 63.23 |
| AVS | N/A | GAT | 55.18 | 48.08 | 54.02 | 56.19 | 55.27 |
| | | HGT | 52.97 | 49.58 | 53.12 | 54.25 | 56.03 |
| | | PNA | 52.78 | 48.72 | 54.19 | 55.45 | 55.06 |
| **Datasets with multiple downstream tasks** | | | | | | | |
| **MAG** | **Venue** | GAT | 44.39 | 47.98 | 47.65 | 51.08 | 51.19 |
| | | HGT | 45.78 | 48.24 | 46.78 | 46.92 | 46.92 |
| | | PNA | 46.60 | 46.25 | 47.36 | 51.59 | 51.59 |
| | **Citation** | GAT | 70.92 | 68.23 | 80.65 | 80.09 | 79.45 |
| | | HGT | 69.95 | 67.73 | 79.31 | 79.05 | 78.96 |
| | | PNA | 70.31 | 65.08 | 77.45 | 77.33 | 77.16 |
| | **Year** | GAT | 54.27 | 54.32 | 54.18 | 56.12 | 35.23 |
| | | HGT | 43.94 | 47.12 | 52.18 | 53.47 | 36.73 |
| | | PNA | 37.85 | 49.75 | 51.26 | 51.68 | 32.39 |
| **Diginetica** | **CTR** | GAT | 50.15 | 68.65 | 50.00 | 73.60 | 73.60 |
| | | HGT | 52.34 | 65.32 | 49.85 | 67.33 | 67.33 |
| | | PNA | 49.88 | 66.43 | 50.15 | 70.15 | 70.15 |
| | **Purchase** | GAT | 4.98 | 7.65 | 15.47 | 37.42 | 37.42 |
| | | HGT | 4.61 | 5.67 | 9.85 | 22.07 | 22.07 |
| | | PNA | 5.33 | 8.05 | 18.52 | 37.58 | 37.58 |
| **Stackexchange** | **Churn** | GAT | 78.04 | 80.96 | 85.43 | 77.77 | 86.45 |
| | | HGT | 78.63 | 76.03 | 85.82 | 87.51 | 86.70 |
| | | PNA | 78.55 | 82.92 | 85.63 | 93.34 | 86.64 |
| | **Upvote** | GAT | 85.96 | 86.54 | 88.53 | 89.00 | 88.53 |
| | | HGT | 84.91 | 85.35 | 88.72 | 86.51 | 88.17 |
| | | PNA | 85.72 | 86.54 | 88.97 | 88.77 | 88.96 |

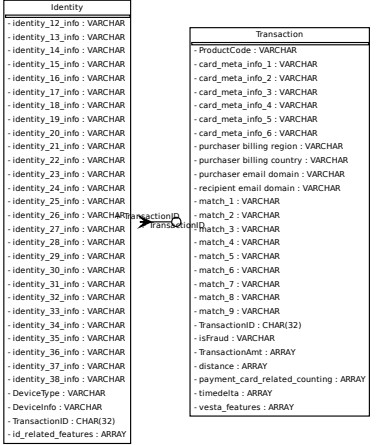

Figure 5: Schema for the AutoG IEEE-CIS dataset

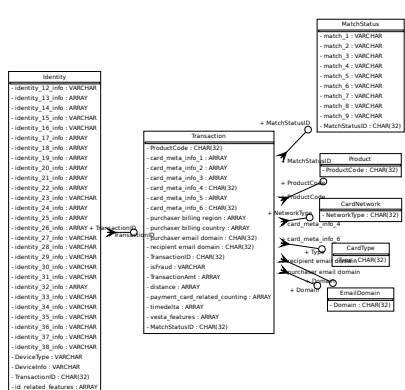

Figure 6: Schema for the expert IEEE-CIS dataset

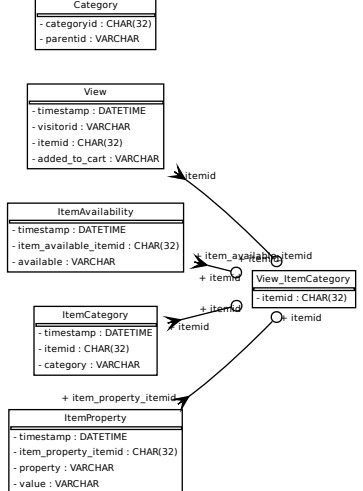

Figure 7: Schema for the AutoG Retail-Rocket dataset

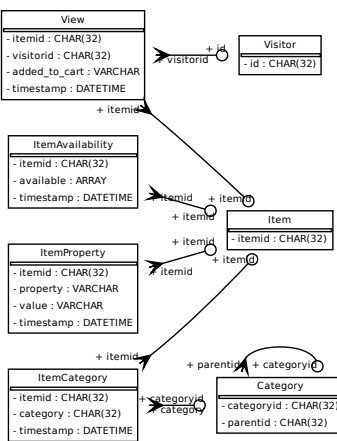

Figure 8: Schema for the expert RetailRocket dataset

### E.3.3 MOVIELENS

Compared to the expert schema, AutoG makes the following variations: 1. It builds a relationship between the user in Tags and Ratings. 2. It identifies the tag column as text type instead of the category type.

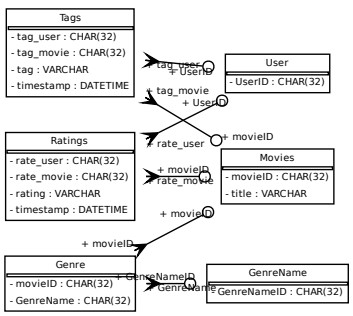

Figure 9: Schema for the AutoG Movielens dataset

Figure 10: Schema for the expert Movielens dataset

### E.3.4 OUTBRAIN

For Outbrain, the difference between AutoG and expert schema lies in utilizing the User dummy table. From the experimental results, we find that such a dummy table presents limited influence on the final performance.

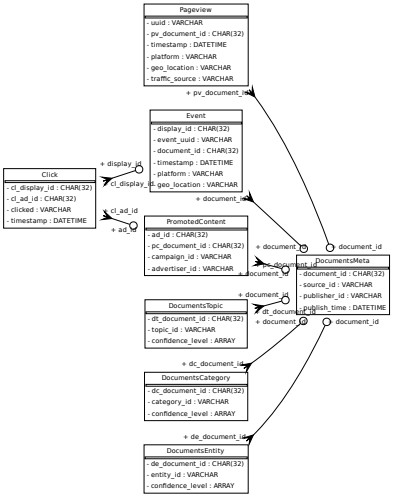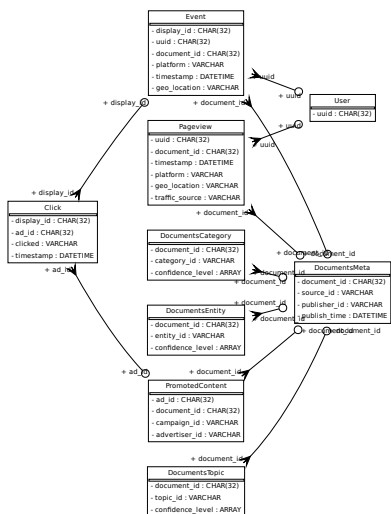

Figure 11: Schema for the AutoG Outbrain dataset

Figure 12: Schema for the expert Outbrain dataset

### E.3.5 AVS

Compared to the expert schema, AutoG doesn't establish a relationship between customers. Instead, it introduces a new dummy table Brand.

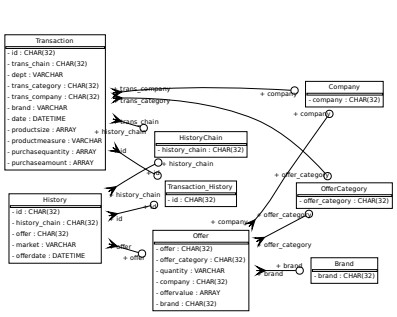

Figure 13: Schema for the AutoG AVS dataset

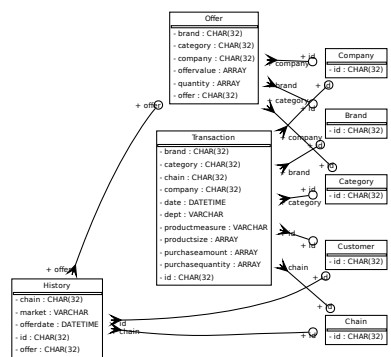

Figure 14: Schema for the expert AVS dataset

### E.3.6 MAG

For MAG, AutoG establishes the same schema for venue prediction and citation prediction task. For the year prediction task, AutoG-S establishes a schema getting around 50% accuracy, while AutoG-A finds that the original schema is the best since there's limited network effect.

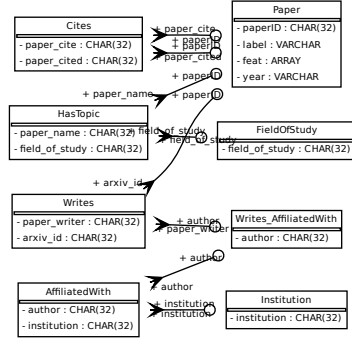

Figure 15: Schema for the AutoG on MAG dataset, venue prediction/citation prediction task

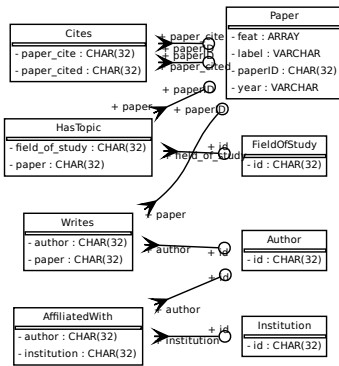

Figure 16: Schema for the expert MAG dataset

### E.3.7 STACKEXCHANGE

For Stackexchange, we change the original userid to corresponding username in the augmented table to improve the augmentation difficulty. As shown below, AutoG doesn't establish this relation after the augmentation. However, from the experimental results, we see that

### E.3.8 DIGINETICA

The expert scheme construction of this dataset has some problems, since some relations don't meet the strict PK-FK relations. AutoG can identify this problem, but since the codebase of 4DBInfer can handle such case, we stick to their implementation. The relation between the CategoryID column is a pitfall. AutoG-S directly generates this relation, and we use AutoG-A to remove this harmful relation.

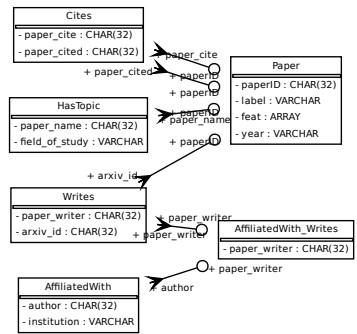

Figure 17: Schema for the AutoG-S on MAG dataset, year prediction task

Figure 18: Schema for the AutoG-A on MAG dataset, year prediction task

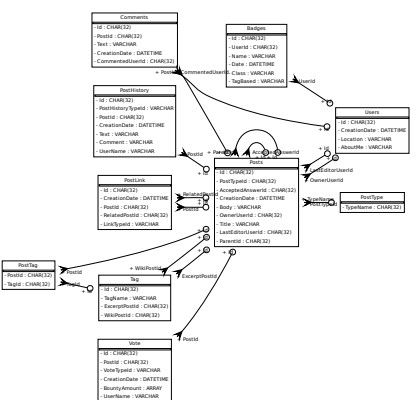

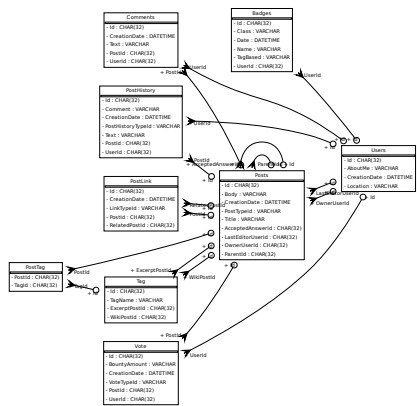

Figure 19: Schema for the AutoG Stackexchange dataset

Figure 20: Schema for the expert Stackexchange dataset

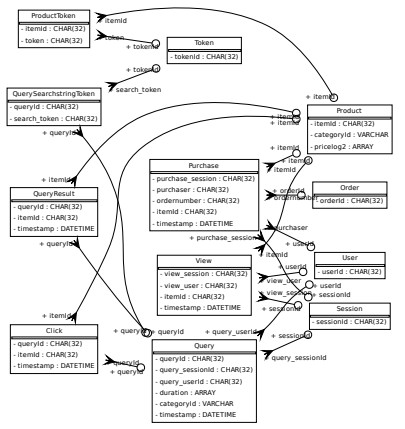

Figure 21: Schema for the AutoG Diginetica dataset

Figure 22: Schema for the expert Diginetica dataset

## E.4 DESIGN OF SYNTHETIC DATASETS

The schema we design for Section 5.3.2 are shown in Figure 23 and Figure 24.

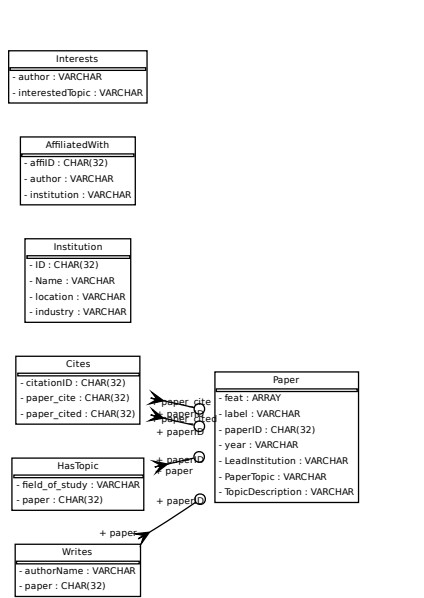

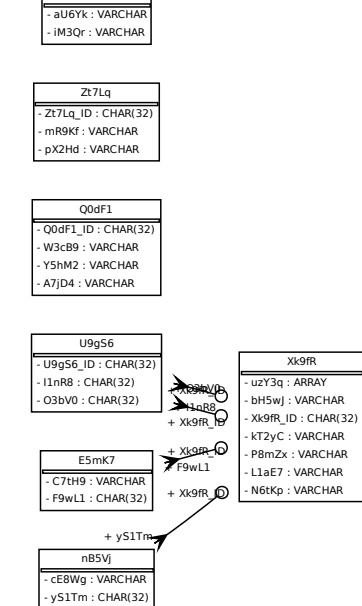

Figure 23: Schema for augmented MAG dataset

Figure 24: Schema for anonymous augmented MAG dataset

