# OpenReview forum: "AutoG: Towards automatic graph construction from tabular data"
_ICLR.cc/2025/Conference — ICLR 2025 Poster_

### Official Review · Reviewer_fBje · 2024-10-31

**Soundness:** 2
**Presentation:** 2
**Contribution:** 2
**Rating:** 6
**Confidence:** 4

**Summary:**

The paper introduces a benchmark for constructing graphs from tabular data as well as an LLM-based method that generates graph schemas without human intervention.

**Strengths:**

(S1) The paper proposes a direction to tackle the interesting problem of transforming tables into graphs.

(S2) The benchmark datasets are a valuable addition to the community.

(S3) The paper performs several experiments to show the effectiveness of the method on various data.

**Weaknesses:**

(W1) The paper does not consider a rich database literature in mining functional dependencies [1,2] and data profiling [3] that automatically extracts dependencies among table columns. These methods can be used to generate more refined schemas that can solve some of the challenges expressed in Section 3.1. This method should be part of the experimental evaluation.

(W2) After having mined uniqueness constraints and functional dependency, one can apply conventional data normalization to further remedy the problems highlighted in challenges C2, and C3. From this standpoint, it is not clear why a mine-and-normal step would not solve most of the issues.

(W4) The benchmark only contains fairly small datasets with, at most 8 tables, for example, PTE has 38 tables (https://relational-data.org/dataset/PTE). There are also several datasets here: https://relational-data.org/

(W5) Another challenge, not specifically addressed by the current paper, is missing values and NULL values. Can the proposed method cope with those? If not, why?

(W6) Clarity: The paper is, in many parts hard to follow. Here are some examples

- Figure 2: It is not clear what flexible design is and what the prompt looks like
- at least two columns as FK and no PK: provide an example
- chain of augmentation: please state what it is
- validation set performance: explain

(W7) 488: How does the method ensure that the test data is not included in the pre-training of the LLM? Do you have access to the training data?

(W8) The paper relies on LLMs that are subject to changes. Does it mean that one has to adapt the method to a different LLM every time?

### Minor

(M1) I encourage the authors to use citations from published sources while in the current draft, several citations are from arXiv

(M2) Typos:

- 133: Treate
- 277: LLM generates / tends → LLMs generate / tend
- 282: single-steply

(M3) In modelling the graph $G={V, E}$ it seems that the nodes are just a set of types as $\mathcal{V}=\bigcup_{v\in V} \mathcal{V}^v$. This definition does not seem standard for Heterogeneous graphs. There should be a set of nodes, each with a type (or a set of types) and a set of edges among the nodes.

[1] Yao, H. and Hamilton, H.J., 2008. Mining functional dependencies from data. *Data Mining and Knowledge Discovery*, *16*, pp.197-219.

[2] Liu, J., Li, J., Liu, C. and Chen, Y., 2010. Discover dependencies from data—a review. *IEEE Transactions on Knowledge and Data Engineering*, *24*(2), pp.251-264.

[3] Abedjan, Z., Golab, L. and Naumann, F., 2015. Profiling relational data: a survey. *The VLDB Journal*, *24*, pp.557-581.

**Questions:**

The authors should carefully address the concern about the relationships with previous database research (W1, W2) and the assumptions of the method (W5). Moreover, the inclusion of a larger, more challenging set of datasets would also be encouraged (W4).

---

> ### Author Response · Authors · 2024-11-20
>
> Thanks for your time and efforts in reviewing.
>
> 1. The paper does not consider a rich database literature in mining functional dependencies [1,2] and data profiling [3] that automatically extracts dependencies among table columns. These methods can be used to generate more refined schemas that can solve some of the challenges expressed in Section 3.1. This method should be part of the experimental evaluation.
>
> Thanks for the great suggestion. We have added the reference you mention in our revision.
>
> In terms of evaluating these methods, it's true that they are very effective for those small-scale datasets like Movielens. However, when they are applied to larger datasets like Diginetica and AVS, it takes significantly long time to detect join discovery. That's because they detect join availability based on statistics and need to look at the whole column distribution. As a comparison, LLM-based methods build upon semantics and can detect based on few-shot samples and statistics.
>
> Moreover, LLM and JTD (embedding-based) methods present both advantages:
> * LLM-based methods are faster, and no need to set a threshold. However, its effectiveness drops with more tables and columns.
> * JTD are effective despite the number of columns. However, threshold need to be set to distinguish joinable and non-joinable columns. In real industrial environment, first filtering possible pairs and then inputing into LLMs is a good practice.
>
> |                  | Movielens | Movielens    | Diginetica | Diginetica   |
> |------------------|-----------|--------------|------------|--------------|
> |                  | Accuracy  | Running time | Accuracy   | Running time |
> | JTD              | 100%      | 75s          | 100%       | 700s         |
> | LLM (CoT)        | 100%      | 15s          | 100%       | 31s          |
> | Cupid [1]        | 100%      | 67s          | N/A        | >3H          |
> | Distribution [2] | 100%      | 45s          | N/A        | >3H          |
>
> [1] Madhavan, Jayant, Philip A. Bernstein, and Erhard Rahm. "Generic schema matching with cupid." vldb. Vol. 1. No. 2001. 2001.
>
> [2] Zhang, M., Hadjieleftheriou, M., Ooi, B. C., Procopiuc, C. M., & Srivastava, D. (2011, June). Automatic discovery of attributes in relational databases. In Proceedings of the 2011 ACM SIGMOD International Conference on Management of data (pp. 109-120).
>
>
>
>
>
>
>
>
> 2. After having mined uniqueness constraints and functional dependency, one can apply conventional data normalization to further remedy the problems highlighted in challenges C2, and C3. From this standpoint, it is not clear why a mine-and-normal step would not solve most of the issues.
>
> Thanks for the great question. We acknowledge that normalization is a closely related concept that is somewhat overlooked in the paper. After investigation, we think challenge 3 can actually be converted to challenge 2 after normalization. However, we want to emphasize the difference between our augmentation goal and normalization in RDB:
>
> The goal of graph construction from relational data to graph is to find what kind of relational information is beneficial to the downstream task (new challenges 1, 2, 3, and 4).
> For example, the objective of challenge 2 is to consider whether the relationship induced by this categorical value is beneficial. This decision needs to consider the semantic relationship between this column and the corresponding downstream tasks and thus cannot be solved by normalization. A student has a category like AI, computer network, system, etc. From the normalization perspective, there is no need to create an augmented table. But from the graph construction perspective, such a student-subject-student relationship is beneficial for predictive tasks.
> As a comparison, the objective of normalization is to minimize data redundancy and improve data integrity. Despite the overlap, data normalization cannot fully solve the graph construction task.
> As discussed in [1], we think the limitation of our method is more that it can handle only simple cases (like when the relation is clear in datasets like MAG, and we also don't consider the case of fuzzy join). Such graph construction is quite challenging and has a long way to go in reality.
>
>
>
> [1] Gan, Quan, et al. "Graph Machine Learning Meets Multi-Table Relational Data." Proceedings of the 30th ACM SIGKDD Conference on Knowledge Discovery and Data Mining. 2024.

---

> ### Author Response · Authors · 2024-11-20
>
> 3. The benchmark only contains fairly small datasets with, at most 8 tables, for example, PTE has 38 tables (https://relational-data.org/dataset/PTE). There are also several datasets here: https://relational-data.org/
>
> Thanks for mentioning these datasets. [1] has done a preliminary study on these datasets, and it shows several pitfalls of adopting these datasets for evaluating Text to graph construction
> 1. Many of these datasets are too easy and simple baselines can get very high performance metrics (100% accuracy or 0 regression loss) that can not well evaluate the effect of different methods.
> 2. The majority of the data predates the era of big data, with only 15 real-world datasets containing more than 10,000 labeled instances.
> 3. Some relations are degenerate and simple outer join and recover all information in the output big table. As a result, graph machine learning doesn't bring clear gain, which makes them not proper for a benchmark evaluating different graph construction methods.
>
> To extend our methods to cases for a large number of tables and columns (which may even surpass the context length of LLMs, as join discovery will be difficult), we consider a variant combining JTD and LLM. We will first use JTD to filter a small number of similar columns and then let LLMs determine the threshold for joinable columns.
>
>
>
> [1] Wang, Minjie, et al. "4DBInfer: A 4D Benchmarking Toolbox for Graph-Centric Predictive Modeling on Relational DBs." Advances in Neural Information Processing Systems 37 (2025).
>
> 4. Another challenge, not specifically addressed by the current paper, is missing values and NULL values. Can the proposed method cope with those? If not, why?
>
> There's missing value and NULL value in the benchmark, for example, in the IEEE-CIS dataset. We address NULL values in two ways:
> 1. If this column is used as a numerical column, we directly fill it using the mean value.
> 2. If this column is used to augment a graph structure relationship (categorical value), this row can be viewed as a special edge type.
>
> In general, our framework can support other missing value-handling methods since these lines of research are orthogonal.
>
>
>
> 5. Clarity: The paper is, in many parts hard to follow. Here are some examples
>
> Figure 2: It is not clear what flexible design is and what the prompt looks like
> at least two columns as FK and no PK: provide an example
> chain of augmentation: please state what it is
> validation set performance: explain
>
> Thank you for bringing these issues to our attention. We will provide clarifications here and revise the original paper accordingly.
>
> 1. **Flexible Design:**  The term "flexible design" refers to modules within our framework that allow for multiple implementations. For instance, in prompt instruction, we can opt for an open-task design (where LLMs can generate any augmentation) or a closed-task design (where LLMs generate augmentations based on predefined actions, leading to improved effectiveness). Examples of prompts, showcasing the following key factors, are provided in Appendix D:
>
>    *  Background of the task
>    *  Statistics of the dataset
>    *  Description of actions
>    *  In-context demonstrations
>
> 2. **Example:** Consider the "Cites" table in the MAG dataset.  Columns in this table are foreign keys referencing the "paperID" column in the "Papers" table. This establishes the **(edge) citation relationship** between the two papers.
>
> 3. **Terminology:** We acknowledge that the current name may be confusing. It simply denotes the series of augmentations generated by LLMs. We will revise this terminology to "a series of augmentations" for clarity.
>
> 4. **Problem Setting:**  In this paper, we assume that graph construction methods have access to labeled training data. We can then partition the training data to create a validation set, allowing us to estimate the performance of trained models on the unseen test set, which is the ultimate goal.
>
>
> 6. 488: How does the method ensure that the test data is not included in the pre-training of the LLM? Do you have access to the training data?
>
> That's a good question. This is a problem for those work using LLMs to simulate the work of data scientists. To address this, in section 5.3.3, we employ a synthetic dataset that is not present in the LLM's training data. The LLM demonstrates effectiveness on this dataset, suggesting its generalizability to most application scenarios.
>
>
> 7. The paper relies on LLMs that are subject to changes. Does it mean that one has to adapt the method to a different LLM every time?
>
> That's a great question. From the experimental results, we find that newly released LLMs and more capable LLMs will achieve stronger performance on this task (Table 5) so there is no need to adapt to LLM updates.

---

> ### Author Response · Authors · 2024-11-20
>
> 8. I encourage the authors to use citations from published sources while in the current draft, several citations are from arXiv
>
> Thanks for pointing this out. For papers that have a published proceeding version, we replaced their citations with the proceeding version.
>
>
> 9. Typos:
>
> 133: Treate
> 277: LLM generates / tends → LLMs generate / tend
> 282: single-steply
>
> Thanks for pointing these out. We have revised all these problems.
>
> 10. In modelling the graph
>  it seems that the nodes are just a set of types as
> . This definition does not seem standard for Heterogeneous graphs. There should be a set of nodes, each with a type (or a set of types) and a set of edges among the nodes.
>
> Thanks for pointing this out. We have revised the definition accordingly (line 129).

---

> > ### Comment · Reviewer_fBje · 2024-11-23
> >
> > I thank the authors for their comprehensive responses and the engaging discussion regarding potential biases. Many of the issues I raised have been addressed effectively, and I have adjusted my score accordingly.
> >
> > However, I still believe that the paper would benefit from a more thorough engagement with the database literature, particularly to contextualize its contributions within existing work.
> >
> > Specifically, data management techniques often prioritize scalability, and I find it hard to fully agree with the statement:
> >
> > > It's true that they are very effective for those small-scale datasets like Movielens. However, when they are applied to larger datasets like Diginetica and AVS, it takes significantly long time to detect join discovery.
> >
> > There are recent works, even incorporating large language models (LLMs), that focus on scalability [2], leverage embeddings [3], or introduce new datasets [1] for evaluation. These examples suggest that scalability challenges can be addressed within the broader scope of data management research.
> >
> > I find the research direction and proposal interesting, but I believe the paper should be positioned more accurately within the context of traditional database research. I remain open to discussing my viewpoint further with the other reviewers.
> >
> > **References:**
> >
> > [1] Koutras, C., Psarakis, K., Siachamis, G., Ionescu, A., Fragkoulis, M., Bonifati, A., & Katsifodimos, A. (2021). Valentine in action: Matching tabular data at scale. Proceedings of the VLDB Endowment, 14(12), 2871-2874.
> >
> > [2] Suhara, Y., Li, J., Li, Y., Zhang, D., Demiralp, Ç., Chen, C., & Tan, W.C. (2022, June). Annotating columns with pre-trained language models. In Proceedings of the 2022 International Conference on Management of Data (pp. 1493-1503).
> >
> > [3] Koutras, C., Fragkoulis, M., Katsifodimos, A., & Lofi, C. (2020, March). REMA: Graph embeddings-based relational schema matching. In EDBT/ICDT Workshops.

---

> > > ### Author Response · Authors · 2024-11-23
> > >
> > > Thank you for the feedback. We have added a discussion of these papers to the related works section. In our study, we use the implementation of Valentine (https://github.com/delftdata/valentine). However, we found that it is significantly slower than LLM-based and LLM embedding-based methods (we applied these methods to every column, though there might be ways to further optimize their efficiency).
> > >
> > > Compared to the datasets typically used in join discovery, the datasets in our study are usually large in the number of rows but small in the number of columns. This difference may explain the performance gap. Heuristic-based methods (statistic-based approaches) require analyzing the entire distribution of column data, while LLM-based methods (either LLMs or LLM embeddings) can approximate the whole column effectively using only a few sampled values—one of the key advantages highlighted in [1].
> > >
> > > For instance, in DeepJoin's implementation, only 512 tokens (equivalent to approximately 10 rows of data) are used to generate column embeddings, which still perform well. This few-shot reasoning capability is one of the reasons why LLMs are particularly useful in data profiling tasks.
> > >
> > > [1] Dong Y, Xiao C, Nozawa T, et al. DeepJoin: Joinable Table Discovery with Pre-trained Language Models[J].

---

> > > ### Author Response · Authors · 2024-11-27
> > >
> > > Dear Reviewer fBje,
> > >
> > > Thank you for taking the time to review our paper.
> > >
> > > We hope our further discussions on data management research have addressed your concerns.
> > >
> > > Since it is approaching the end of the discussion period, if you have any further questions or feedback, please don’t hesitate to let us know!
> > >
> > > Best regards,
> > >
> > > Authors

---

> > > > ### Comment · Reviewer_fBje · 2024-11-28
> > > >
> > > > Thanks again for the discussion on this topic.
> > > >
> > > > I am wondering, if the problem is the number of rows, one can simply sample a few rows (good question on how do that and preserve the correct joins) and then apply traditional data management techniques or DeepJoin.
> > > > Is that not a viable solution?

---

> > > > > ### Author Response · Authors · 2024-11-28
> > > > >
> > > > > Thank you for your feedback! We are happy to provide further clarification.
> > > > >
> > > > >
> > > > > First, It’s possible that a traditional join method could perform well if we first sort the column values (e.g., alphabetically) and then sample the first \(k\) columns. This approach might detect joinable relations if the column distributions are sufficiently similar. However, as benchmarked in [1], the performance of DeepJoin (referred to as the "JTD" baseline in our paper) likely represents the upper bound for these traditional methods. In hard discovery cases (no relation is pre-given), our system also relies on the output results of Deepjoin, and LLM determines the threshold for join discovery. Our framework is flexible and any join discovery method result can be used as input.
> > > > >
> > > > >
> > > > > Second, We also want to clarify the objective of join discovery when applying graph machine learning models to relational data. As discussed in [2], this involves three levels of relationships:
> > > > >
> > > > > 1. **Primary Key-Foreign Key (PK-FK) Relationships**: These relate existing primary keys to non-key columns.
> > > > > 2. **Foreign Key-Foreign Key (FK-FK) Relationships**: These connect two non-key columns.
> > > > > 3. **Foreign Key Relationships**: These involve a single non-key column and may include "self-joins" within the same table.
> > > > >
> > > > > In addition to these, our paper highlights a fourth level:
> > > > >
> > > > > 4. **Benign Relations for Downstream Tasks**: Not all relations are beneficial for downstream tasks. Some can even be detrimental, as seen in cases like MAG-year and IEEE-CIS.
> > > > >
> > > > > We believe **traditional data management tools can only address the first two cases**. It will be very challenging to tackle the latter two with traditional management tools. These challenges are at the heart of graph construction for relational data.
> > > > >
> > > > > In this paper, we make a preliminary attempt to address these challenges by leveraging large language models (LLMs) as "graph data scientists" to identify benign relationships. We supply LLMs with column statistics, and optionally graph statistics or task performance data, enabling them to reason about the semantics of the relationships. For instance, they might assess the "paper-paper" citation relationship and determine its utility for specific downstream tasks like predicting a paper’s category (stronger correlation, higher homophily) or publication year (weaker correlation).
> > > > >
> > > > > With this design, we demonstrate that LLMs exhibit preliminary reasoning capabilities previously achievable only by human experts.
> > > > >
> > > > >
> > > > > [1] Koutras, Christos, et al. "OmniMatch: Effective Self-Supervised Any-Join Discovery in Tabular Data Repositories." *arXiv preprint arXiv:2403.07653* (2024).
> > > > > [2] Gan, Quan, et al. "Graph Machine Learning Meets Multi-Table Relational Data." *Proceedings of the 30th ACM SIGKDD Conference on Knowledge Discovery and Data Mining.* 2024.

---

> > > > > > ### Comment · Reviewer_fBje · 2024-11-28
> > > > > >
> > > > > > Thanks for the extensive answer and discussion.
> > > > > >
> > > > > > > First, It’s possible that a traditional join method could perform well if we first sort the column values (e.g., alphabetically) and then sample the first (k) columns.
> > > > > >
> > > > > > The previous answer was referring to sampling rows not columns as the main bottleneck of relational approaches (e.g., JTD) seemed the number of rows.
> > > > > >
> > > > > > > We believe traditional data management tools can only address the first two cases. It will be very challenging to tackle the latter two with traditional management tools. These challenges are at the heart of graph construction for relational data.
> > > > > >
> > > > > > I can relate to the point of constructing a graph functional to a certain downstream task, and I believe the point 3 and 4 are indeed important to achieve such a goal.

---

> > > > > > > ### Author Response · Authors · 2024-11-28
> > > > > > >
> > > > > > > Thank you for the additional feedback!
> > > > > > >
> > > > > > > > The previous response referred to sampling rows, not columns, as the main bottleneck of relational approaches (e.g., JTD) seems to be the number of rows.
> > > > > > >
> > > > > > > The primary goal of join discovery methods is to scale efficiently with both rows and columns. Traditional heuristic-based methods, such as Cupid, scale their runtime quickly relative to the number of rows. Sampling rows can help mitigate this issue if we can identify a representative subset. Techniques inspired by active learning or coreset selection could be employed to improve sampling effectiveness, but it's not trivial to design the metrics.
> > > > > > >
> > > > > > > Methods based on language model embeddings, such as JTD, scale well with respect to both rows and columns. LLM-based methods, despite their efficiency, are constrained by the context length of the language model. For self-join discovery, we can input subsets of a table, but join discovery—being a pairwise task—requires external support like JTD when working with lots of columns (for datasets in this paper, pure LLM seems the best choice).
> > > > > > >
> > > > > > > Overall, we believe that the most effective approach may be a hybrid of JTD and LLM, combining their respective strengths to tackle these challenges (in this paper, their effectiveness should be close to pure LLM-based methods).

---

### Official Review · Reviewer_8qV7 · 2024-11-03

**Soundness:** 2
**Presentation:** 3
**Contribution:** 2
**Rating:** 6
**Confidence:** 3

**Summary:**

This paper focuses on the impact of automatic graph construction from input tabular data on downstream Graph Machine Learning (GML) tasks. It introduces a benchmark for evaluating graph construction and proposes AutoG, an automated graph construction method.

**Strengths:**

1. In the field of GML methods for tabular data, this paper is the first to concern graph construction evaluation, and introduces a benchmark for it.
2. The authors identify five key challenges in converting tabular data into graphs and proposed AutoG, an agent-based approach. They design actions for the large language model (LLM) to utilize its prior knowledge for augmenting graph construction. Additionally, they implement a feedback mechanism to calibrate the LLM's output based on the validation performance of downstream GML tasks.
3. The paper is well-structured, clearly expressed, and includes detailed information.

**Weaknesses:**

My main concern is with the "metrics used in the proposed benchmark for evaluating the conversion of tabular data into graphs (T2G)." As stated in line 216, the authors use the performance of fixed GML models (RGCN, RGAT) trained on the generated graphs for quantitative evaluation. However, as a benchmark for T2G, it should be independent of specific downstream GML methods. Thus, the evaluation is limited to these two models and lacks generalizability to all GML models. Until it is demonstrated that the conclusions from RGCN and RGAT apply to all GML models, both the benchmark and the experimental findings in this paper remain constrained.

**Questions:**

1. Can you prove that the conclusions from RGCN and RGAT apply to all GML models?
2. Have you considered proposing a better metric to evaluate the constructed graph itself? Furthermore, if the constructed graph is closely tied to specific downstream tasks, does that suggest that T2G might be more suitable as a data augmentation module to be explored with specific GML models, rather than as a standalone benchmark?

---

> ### Author Response · Authors · 2024-11-20
>
> Thanks for your time and efforts in reviewing.
>
> 1/2. My main concern is with the "metrics used in the proposed benchmark for evaluating the conversion of tabular data into graphs (T2G)." As stated in line 216, the authors use the performance of fixed GML models (RGCN, RGAT) trained on the generated graphs for quantitative evaluation. However, as a benchmark for T2G, it should be independent of specific downstream GML methods. Thus, the evaluation is limited to these two models and lacks generalizability to all GML models. Until it is demonstrated that the conclusions from RGCN and RGAT apply to all GML models, both the benchmark and the experimental findings in this paper remain constrained. Can you prove that the conclusions from RGCN and RGAT apply to all GML models?
>
> That's a great question. We think the conclusions shown in our paper can extend to any GML models conducting early fusion [1], which refers to typical message-passing or self-attention-based models like RGCN, RGAT, and graph transformers. As evidence, the performance disparity shown in [1] among different "early fusion" models is very small.
>
> |                        | AVS   | Diginetica (CTR) | MAG (venue) | MAG (cite) |
> |------------------------|-------|------------------|-------------|------------|
> | DeepFM on joined table | 56.20 | 50.57            | 28.19       | 45.06      |
> | HGT                    | 57.03 | 67.33            | 46.92       | 78.96      |
> | RGCN                   | 56.53 | 75.07            | 49.36       | 80.65      |
> | RGAT                   | 56.38 | 73.20            | 51.19       | 79.40      |
> | PNA                    | 56.08 | 70.11            | 51.59       | 77.16      |
>
>
>
>
>
> [1] Wang, Minjie, et al. "4DBInfer: A 4D Benchmarking Toolbox for Graph-Centric Predictive Modeling on Relational DBs." Advances in Neural Information Processing Systems 37 (2025).
>
>
>
>
>
> 3. Have you considered proposing a better metric to evaluate the constructed graph itself? Furthermore, if the constructed graph is closely tied to specific downstream tasks, does that suggest that T2G might be more suitable as a data augmentation module to be explored with specific GML models, rather than as a standalone benchmark?
>
> That's a good suggestion. We attempt some related graph properties to reflect the generated graph properties, such as the homophily ratio. However, as discussed in Section 4, we find that these metrics cannot well reflect the final goal, which is the downstream task performance. In this sense,  we follow the philosohpy of [2], and adopt model as anchors to reflect the quality of constructed graphs.
>
> We acknowledge that the constructed graph is tied to downstream tasks, and we think it's an important property of this benchmark. Unlike [1] which adopts the same graph for each downstream task, we find that adaptive graph construction is critical to good downstream task performance. As a result, we design our benchmark to be a **data-centric** benchmark and the core is to test different methods generating the data structure while different GML models are adopted as anchors to reflect the quality of graphs.
>
> [1] Wang, Minjie, et al. "4DBInfer: A 4D Benchmarking Toolbox for Graph-Centric Predictive Modeling on Relational DBs." Advances in Neural Information Processing Systems 37 (2025).
>
> [2] You, Jiaxuan, Zhitao Ying, and Jure Leskovec. "Design space for graph neural networks." Advances in Neural Information Processing Systems 33 (2020): 17009-17021.

---

> > ### Comment · Reviewer_8qV7 · 2024-11-22
> >
> > Thank you for the thoughtful responses.
> >
> > However, I still have the following concern and suggestion: Since Graph Transformer and PNA are mentioned, would it be more comprehensive to evaluate T2G as a benchmark by including the performance of a set of GML models, such as GCN, GAT, Graph Transformer, and PNA? For instance, [2] utilized a set of 12 GNN designs as anchors for GNN task similarity evaluation. This approach might provide a broader and more balanced assessment than relying solely on GCN/GAT.

---

> ### Author Response · Authors · 2024-11-22
>
> Thank you for the feedback.
> First, we use the relational version of graph transformer (HGT) and PNA (R-PNA), and GNNs designed for homogeneous graphs perform consistently worse than those models designed for heterogeneous graphs [1].
> The 12 GNNs in [2] are actually unified model architectures with varying hyperparameter settings designed to evaluate task similarity for homogeneous graphs. In contrast, our goal is to assess graph quality for heterogeneous graphs. To achieve this, we also perform a hyperparameter search to identify the best configuration, which should effectively reflect the quality of the graph.
>
> [1] Lv, Qingsong, et al. "Are we really making much progress? revisiting, benchmarking and refining heterogeneous graph neural networks." Proceedings of the 27th ACM SIGKDD conference on knowledge discovery & data mining. 2021.

---

> > ### Comment · Reviewer_8qV7 · 2024-11-23
> >
> > Thank you for your response. I may not have expressed myself clearly. My point is that, similar to [2], which evaluates combinations of different GNN Design Spaces and GNN Task Spaces, your work involves combinations of different T2G methods, heterogeneous GML models, and downstream tasks. Would it be possible to consider using the "controlled random search" approach described in Section 6 of [2] to sample and evaluate combinations in your scenario?

---

> > > ### Author Response · Authors · 2024-11-23
> > >
> > > Thank you for the clarification. In our setting, the downstream task is the main objective. Combining T2G methods with heterogeneous graph models is feasible, and the conclusions should generally hold. For instance, we conducted a case study on the MAG dataset using the venue prediction task (with a limited grid search for HGT and RPNA). The relative relationships between the average performance of models and the performance of individual models were largely consistent. For example, when all relationships are utilized—such as in AutoG—it consistently achieves the best results.
> > >
> > > We also acknowledge that using the average performance across a set of models can serve as a more robust metric. This comes at the cost of requiring thorough hyperparameter tuning for each model in every task and every schema. As a result, using representative model performance (RGCN and RGAT) is a cost-effective way to demonstrate the superiority of AutoG's schema.
> > >
> > >
> > > |      | Original | JTD   | TabGNN | AutoG |
> > > |------|----------|-------|--------|-------|
> > > | RGCN | 46.97    | 46.26 | 42.84  | 49.88 |
> > > | RGAT | 47.98    | 47.65 | 44.39  | 51.08 |
> > > | HGT  | 48.24    | 46.78 | 45.78  | 49.84 |
> > > | RPNA | 46.25    | 47.36 | 46.60  | 51.59 |
> > > | Avg  |    47.36 | 47.01 |  44.90 | 50.59 |

---

> > > > ### Comment · Reviewer_8qV7 · 2024-11-24
> > > >
> > > > We appreciate the additional experiments provided by the authors.
> > > >
> > > > However, I would like to clarify that my concern **is not whether AutoG outperforms other T2G methods but whether the proposed Benchmark Evaluation for T2G methods is reasonable**.
> > > >
> > > > 1. First, let's consider the ranking of each T2G method for the same GML model based on your added experiments:
> > > >
> > > > | Model  | Original | JTD | TabGNN | AutoG |
> > > > |:------:|:--------:|:---:|:------:|:-----:|
> > > > | RGCN   |    2    |  3  |   4    |   1   |
> > > > | RGAT   |    2     |  3  |   4    |   1   |
> > > > | HGT    |    2     |  3  |   4    |   1  |
> > > > | RPNA   |    4     |  2  |   3    |   1   |
> > > >
> > > > It becomes evident that **even for the same downstream task, the ranking of different T2G methods does not remain consistent across different GML models**. For example, for RPNA, +JTD outperforms +Original, but for RGCN, RGAT, and HGT, +Original consistently ranks higher than +JTD.
> > > >
> > > > 2. Second, we can convert your added experiments into rankings of each GML model for the same T2G method:
> > > >
> > > > | Model  | RGCN | RGAT | HGT | RPNA |
> > > > |:------:|:----:|:----:|:---:|:----:|
> > > > | Original |  3   |  2   |  1  |  4   |
> > > > | JTD      |  4   |  1   |  3  |  2   |
> > > > | TabGNN   |  4   |  3   |  2  |  1   |
> > > > | AutoG    |  3   |  2   |  4  |  1   |
> > > >
> > > > It is obvious that **RGAT/RGCN do not always perform the best and cannot be considere representative**. In fact, only RGAT ranks first for +JTD. For the other three T2G methods (including AutoG), HGT and RPNA perform better for the same T2G.
> > > >
> > > > These inconsistencies confirm my concern that **using the performance of a T2G method on a single GML model to represent its performance across all GML models is not appropriate**. I still believe that modifying the Benchmark Evaluation approach, as I previously suggested, i.e., **by using the average performance across various combinations of GML models and downstream tasks**, is necessary.

---

> > > > > ### Author Response · Authors · 2024-11-24
> > > > >
> > > > > Thanks for the active engagement with us and continuous feedback. We really appreciate it. Since the metric we used is to evaluate the quality of the graph with respect to each task and each method below, we refer to it as an “individual” metric while we refer to the one suggested by you as an “average” metric.  We think that via our discussions, you indeed suggest a metric (“average”) that is complementary to the current one we use (“individual”).  Let us elaborate on why.
> > > > >
> > > > > The **average** metric is an average across datasets and models. It is important since it can reflect an overall assessment of the graph quality. The “individual” metric is also important. First, it can give a fine-grained assessment of the graph quality for different tasks and different methods. This evaluation is crucial if we are interested in some specific models or specific tasks. Second, given the wide adoption of R-GCN(R-SAGE)/R-GAT as the main backbone models for various relational deep learning libraries [1,2] (when they talk about GNN, they usually refer to R-GCN). It can reflect the stability of graph construction methods on these mainstream models.  From the ranking tables you provided, the performance is indeed unstable across models (maybe tasks, too). Therefore both the “average” and “individual” are important and complementary since they provide graph quality from different perspectives.
> > > > >
> > > > > Base on the above, we agree with you that we should provide both in our paper. We are running experiments and will send a response and revision once we have done the experiments. However, given that AutoG almost consistently outperforms all the baseline methods (in the main table of the paper and Table 1) given different backbone models, we believe the main conclusion of our paper that AutoG outperforms other automatic graph construction methods is consistent given both **individual** and **average** metric.
> > > > >
> > > > > We really enjoy the interactions with you which make our work more solid. We are looking forward to your feedback.
> > > > >
> > > > > [1] Robinson, Joshua, et al. “Relbench: A benchmark for deep learning on relational databases.” arXiv preprint arXiv:2407.20060 (2024).
> > > > >
> > > > > [2] Zheng, Da, et al. “GraphStorm: All-in-one graph machine learning framework for industry applications.” Proceedings of the 30th ACM SIGKDD Conference on Knowledge Discovery and Data Mining. 2024.

---

> > > > > > ### Comment · Reviewer_8qV7 · 2024-11-25
> > > > > >
> > > > > > I appreciate the authors’ active response and discussion. I agree that the "individual metric" is important. In fact, I highlighted this in my initial review: "Does that suggest that T2G might be more suitable as a data augmentation module to be explored with specific GML models, rather than as a standalone benchmark?"
> > > > > >
> > > > > > My concern has always been centered on the first contribution claimed in your paper: "First, we introduce a benchmark to formalize and evaluate graph construction methods."
> > > > > > Since one of the key contributions of the paper is the "T2G Benchmark," I believe it is necessary to provide a comprehensive evaluation that includes a variety of GML models and downstream tasks. Such an approach would not only strengthen the validity of the benchmark but might also uncover more interesting insights.

---

> > > > > > > ### Author Response · Authors · 2024-11-26
> > > > > > >
> > > > > > > Thank you for your active engagement and continuous feedback. We have conducted additional experiments to evaluate the average performance of model sets and generated updated rankings. In cases such as Movielens and MAG year, we observed differences between evaluations using RGCN and model sets, highlighting that using average performance provides a more reasonable assessment. Furthermore, we found that hyperparameters play a crucial role in determining final performance. Given that grid search does not always guarantee the optimal hyperparameter settings for each model, by using average performance, we can mitigate the bias introduced by hyperparameter selection.
> > > > > > >
> > > > > > > |                   | TabGNN | Original | JTD   | AutoG | Expert |
> > > > > > > |-------------------|--------|----------|-------|-------|--------|
> > > > > > > | IEEE-CIS fraud    | 75.34  | 88.88    | 88.88 | 90.11 | 89.32  |
> > > > > > > | RR CVR            | 82.75  | 49.85    | 50.58 | 82.99 | 84.19  |
> > > > > > > | Movielens ratings | 59.37  | 59.14    | 63.34 | 69.69 | 69.69  |
> > > > > > > | Outbrains ctr     | 62.44  | 52.37    | 52.68 | 61.62 | 63.06  |
> > > > > > > | AVS repeat        | 53.85  | 48.81    | 53.65 | 54.40 | 55.36  |
> > > > > > > | MAG venue         | 44.90  | 47.18    | 47.19 | 49.87 | 49.94  |
> > > > > > > | MAG citation      | 70.46  | 66.58    | 79.73 | 79.33 | 79.11  |
> > > > > > > | MAG year          | 47.21  | 51.32    | 52.67 | 53.84 | 34.93  |
> > > > > > > | DIG CTR           | 50.59  | 67.21    | 50.01 | 70.43 | 71.54  |
> > > > > > > | DIG Purchase      | 4.98   | 7.27     | 14.83 | 31.99 | 33.50  |
> > > > > > > | STE churn         | 78.37  | 75.82    | 85.62 | 86.00 | 86.34  |
> > > > > > > | STE upvote        | 85.47  | 88.78    | 88.71 | 88.50 | 88.57  |
> > > > > > >
> > > > > > >
> > > > > > > |         | XGBoost | DeepFM | TabGNN | Original | JTD | AutoG | Expert |
> > > > > > > |---------|---------|--------|--------|----------|-----|-------|--------|
> > > > > > > | Ranking | 5.8     | 5.5    | 4.5    | 4.5      | 3.4 | 2.3   | 1.9    |
> > > > > > >
> > > > > > >
> > > > > > > We completely agree that the previous evaluation method lacks the rigor expected for a benchmark evaluation, which undermines the strength of the benchmark claims in our paper. In fact, the primary focus of this paper is on the methodological design of automating graph construction using LLMs, rather than on establishing a benchmark. Consequently, the paper was not submitted under the benchmark category. We acknowledge the issue with the original wording and have revised it to "proposed datasets" to more accurately reflect the paper's contribution.

---

> > > > > > > > ### Comment · Reviewer_8qV7 · 2024-11-26
> > > > > > > >
> > > > > > > > Thank you for your thoughtful responses. I hope our discussion has contributed to improving your paper. I have raised my score by one point.

---

> > > > > > > > > ### Author Response · Authors · 2024-11-26
> > > > > > > > >
> > > > > > > > > Thank you for your proactive feedback and constructive suggestions, which have helped us address the issues in the paper.

---

### Official Review · Reviewer_4sTF · 2024-11-04

**Soundness:** 3
**Presentation:** 3
**Contribution:** 3
**Rating:** 6
**Confidence:** 3

**Summary:**

The authors provide a method for automatically converting relational tables into a knowledge graph. They forego manual engineering by using an LLM approach. First, the LLM is instructed to create a schema, restricted by calling certain functions. Then, an oracle is used to provide feedback for this schema generation. That feedback is then again used by the generator.

**Strengths:**

* The general observation is an important one: the conversion of tabular data into graph form cannot be taken for granted. Existing graph benchmarks based on tables avoid the hard cases.
* The setup with an LLM to generate candidates is pretty nice. Restricting its generation freedom using function calls is a good idea as well.

**Weaknesses:**

* It remains unclear how well this method performs on the long tail. The evaluation is averaging over many cases, but the results might be dwarfed by very common ones.
* The paper only considers very well behaving rectangular tables (relational database style), with column names and all. Even the data types are given. There is also a lot of web tables around with large datasets available. One could certainly question whether te chosen setting is realistic, now.
* The datasets are pretty small.
* Citations for relational graph learning are to 2023< papers, while this has been studies for 10+ years.
* It is not clearly argued why relational graphs are most suitable. Other formalisms like hyper-relational ones might be more appropriate.
* I suggest removing the claim of surpassing human experts. It is not clear whether all schemas from the experts are  such that they were made as an ideal schema for a GNN architecture / the specific task.

minor:
*  treate each ->  treated each

**Questions:**

* Recently, Alivanistos, et al. (2024) presented "The Effect of Knowledge Graph Schema on Classifying Future Research Suggestions". That work suggests that the classification performance depends on the chosen schema. Is this the same you observed in the current work? Should one consider existing schemas or let the model figure this creation out as well?
* Much of the idea is clearly supported by argument, the choice of the oracle as a discriminator and the heuristics one gets out of that are rather arbitrary. It is not clear whether these are the best options. How do you know?
* You only use RGCN and GAT, both of which are rather GNN based learning methods. Would you expect other methods to behave similarly? Why?

---

> ### Author Response · Authors · 2024-11-20
>
> Thanks for your time and efforts in reviewing.
> 1. It remains unclear how well this method performs on the long tail. The evaluation is averaging over many cases, but the results might be dwarfed by very common ones.
>
> Thanks for the great question. The "long tail" node is usually defined as those nodes with lower degrees. We do a case study on the MAG dataset. We first split nodes into 5 bins according to their degrees.
>
> |            | MAG (venue) |
> |------------|-------------|
> | Overall    | 47.58       |
> | First bin  | 47.57       |
> | Second bin | 49.02       |
> | Third bin  | 47.18       |
> | Fourth bin | 45.71       |
> | Fifth bin  | 48.42       |
> | DeepFM     | 28.19       |
>
> We find that actually the boost brought by graph construction distributes across nodes with different degrees in a relatively uniform manner.
>
>
> 2. The paper only considers very well behaving rectangular tables (relational database style), with column names and all. Even the data types are given. There is also a lot of web tables around with large datasets available. One could certainly question whether te chosen setting is realistic, now.
>
> Thanks for the great question. You make a good point that one property of LLM-based methods of automatic data engineering is the dependency of column semantics, which is also demonstrated in [1,2].
>
> The current setting is inspired by [3], where we assume several tables are extracted from data lakes with missing or incomplete relations for downstream predictive tasks. One limitation of the selected dataset may be the limited size of columns and the focus on exact join instead of fuzzy join/semantic join. As a result, despite the simplicity of our setting, it reflects real industrial scenarios well.
>
> [1] Hollmann, Noah, Samuel Müller, and Frank Hutter. "Large language models for automated data science: Introducing caafe for context-aware automated feature engineering." Advances in Neural Information Processing Systems 36 (2024).
>
> [2] Han, Sungwon, et al. "Large Language Models Can Automatically Engineer Features for Few-Shot Tabular Learning." Forty-first International Conference on Machine Learning.
>
> [3] Gan, Quan, et al. "Graph Machine Learning Meets Multi-Table Relational Data." Proceedings of the 30th ACM SIGKDD Conference on Knowledge Discovery and Data Mining. 2024.
>
>
> 3. The datasets are pretty small.
>
> The number of tables and columns in our datasets is relatively small, while the size of our dataset is not. One reason for this is the absence of datasets with a large number of tables and columns for the evaluation of GML models. As shown in [1], which conducts a preliminary study on datasets from https://relational-data.org/. The results show that either those datasets are too easy or do not present valuable structures for evaluating GML models. As a result, we mainly adopt datasets from [1].
>
> Moreover, to extend the potential of our pipeline to large-scale datasets, we further demonstrate the following extension: we consider a variant combining LLM and JTD. For tables with many columns and tables, we first filter columns with high similarity and feed them to LLM. In this way, we can combine the advantages of the two methods with the cost of more computation overhead.
>
>
> [1] Wang, Minjie, et al. "4DBInfer: A 4D Benchmarking Toolbox for Graph-Centric Predictive Modeling on Relational DBs." Advances in Neural Information Processing Systems 37 (2025).
>
>
>
> 4. Citations for relational graph learning are to 2023< papers, while this has been studies for 10+ years.
>
> Thanks for pointing this out. We have followed [1, 2] and added a discussion in Appendix C.
>
>
> [1] Zhou, Jie, et al. "Graph neural networks: A review of methods and applications." AI open 1 (2020): 57-81.
>
> [2] Yang, Carl, et al. "Heterogeneous network representation learning: A unified framework with survey and benchmark." IEEE Transactions on Knowledge and Data Engineering 34.10 (2020): 4854-4873.

---

> ### Author Response · Authors · 2024-11-20
>
> 5. It is not clearly argued why relational graphs are most suitable. Other formalisms like hyper-relational ones might be more appropriate.
>
> That's a great question!  While hypergraphs have shown promise for representing single tabular datasets[1,2], the most effective way to apply graph-structured machine learning to complex, multi-tabular data remains an active area of research.  For our work, we've chosen to utilize a message-passing GNN pipeline, the dominant approach in industry-level benchmarks [3,4], as it offers superior scalability and flexibility for capturing the intricate relationships within these datasets. This allows us to build upon an established foundation, ensuring comparability and reproducibility of our results.
>
>
> [1] Chen, Pei, et al. "HYTREL: Hypergraph-enhanced tabular data representation learning." Advances in Neural Information Processing Systems 36 (2024).
>
> [2] Du, Kounianhua, et al. "Learning enhanced representation for tabular data via neighborhood propagation." Advances in Neural Information Processing Systems 35 (2022): 16373-16384.
>
> [3] Wang, Minjie, et al. "4DBInfer: A 4D Benchmarking Toolbox for Graph-Centric Predictive Modeling on Relational DBs." Advances in Neural Information Processing Systems 37 (2025).
>
> [4] Robinson, Joshua, et al. "Relbench: A benchmark for deep learning on relational databases." arXiv preprint arXiv:2407.20060 (2024).
>
>
>
>
> 6. I suggest removing the claim of surpassing human experts. It is not clear whether all schemas from the experts are such that they were made as an ideal schema for a GNN architecture / the specific task.
>
> You're right to point that out. We've removed the claim and appreciate you bringing it to our attention.
>
> It's true that schemas designed by human experts might not be optimized for every specific task. For instance, on the MAG dataset, the expert-designed schema didn't perform well for year prediction.  This highlights a key advantage of our LLM-based approach: it can automatically generate task-specific schemas, saving significant time and effort compared to manual design. This adaptability is crucial, especially when dealing with diverse tasks and datasets.
>
>
> 7. Recently, Alivanistos, et al. (2024) presented "The Effect of Knowledge Graph Schema on Classifying Future Research Suggestions". That work suggests that the classification performance depends on the chosen schema. Is this the same you observed in the current work? Should one consider existing schemas or let the model figure this creation out as well?
>
> The work you mention is about the creation of a knowledge graph from text using some fixed schemas, and the result demonstrates that they greatly influence the final downstream task performance. Processing tabular data often involves a two-stage pipeline. The first stage focuses on transforming raw data into a structured format, while the second stage involves constructing graphs from this structured data to facilitate analysis. We think the work you mention is related to the first stage.
> While some research concentrates on the first stage [1], our work specifically addresses the second stage:  constructing graphs from pre-structured data.
> While we haven't explicitly explored the schema influence to the same extent, we agree that schema design is critical. An ill-defined schema can lead to information loss and hinder effective analysis. For instance, existing schemas designed for scientific articles might not be suitable for more diverse data formats.
>
> Regarding LLMs in the first stage, we've observed challenges with both code generation (LLMs generate pre-processing code, often resulting in non-executable code for complex cases) and direct structured output (LLMs directly output json output, limited to shorter inputs).  While LLMs are powerful tools, human expertise remains crucial for reliable and comprehensive data structuring at this stage. As a result, we think a viable approach is to adopt LLMs together with manual post-processing.
>
>
>
> [1] Wu, Haolun, et al. "Structured Entity Extraction Using Large Language Models." arXiv preprint arXiv:2402.04437 (2024).

---

> ### Author Response · Authors · 2024-11-20
>
> 8. Much of the idea is clearly supported by argument, the choice of the oracle as a discriminator and the heuristics one gets out of that are rather arbitrary. It is not clear whether these are the best options. How do you know?
>
> We appreciate your acknowledgment of the reasonableness of our design choices. Regarding the use of a GNN as an oracle, our primary goal is to assess the quality of the generated graphs. While oracles exist for generated molecular graphs, utilizing metrics like chemical properties [2], evaluating the quality of industrial graphs presents a more significant challenge. Therefore, we adopt the approach from [1], which employs a model as an anchor for quality determination. The underlying principle is that superior performance of the trained model on the visible validation set indicates a higher quality generated graph.
>
> In the best-case scenario, given the vast search space of potential graph structures, definitively identifying the optimal graph is difficult. Consequently, to evaluate the effectiveness of our generated graph, we use the design from [3] as a benchmark. If our generated graph leads to improved performance on the downstream task, we consider it a successful design.
>
>
> [1] You, Jiaxuan, Zhitao Ying, and Jure Leskovec. "Design space for graph neural networks." Advances in Neural Information Processing Systems 33 (2020): 17009-17021.
>
> [2] Wang, Haorui, et al. "Efficient evolutionary search over chemical space with large language models." arXiv preprint arXiv:2406.16976 (2024).
>
> [3] Wang, Minjie, et al. "4DBInfer: A 4D Benchmarking Toolbox for Graph-Centric Predictive Modeling on Relational DBs." Advances in Neural Information Processing Systems 37 (2025).
>
>
>
> 9.You only use RGCN and GAT, both of which are rather GNN based learning methods. Would you expect other methods to behave similarly? Why?
>
> We expect other GML methods behave similarly. For example, [1] evaluates more GML models like graph transformer and PNA, and the conclusion is consistent for different GML models.
>
> However, as shown in our experiment, there are some other possible model architectures like tabular models such as XGBoost and DeepFM, they can not enjoy the improvement brought by better graph design since they are graph-agnostic. For feature synthesis method like DFS, we also don't think the behavior will be similar, since they first conduct a feature synthesis based on relations, which is different from the parametric aggregation adopted in GNN-based and graph transformer-based models.
>
>
> [1] Wang, Minjie, et al. "4DBInfer: A 4D Benchmarking Toolbox for Graph-Centric Predictive Modeling on Relational DBs." Advances in Neural Information Processing Systems 37 (2025).

---

> ### Comment · Reviewer_4sTF · 2024-11-20
> **Thank you for your responses.**
>
> Thank you for your answers. A couple of aspect remain worrisome to me.
>
> Regarding W1: What does worry me a bit, is that you only look at venues in the graph, which I would expect to actually not have all that much variation. What do you obtain for other types? Overall, I think it would be useful to have some more extensive results on the long tail investigation in the paper. This is actually a nice finding as such.
>
> Regarding W5: your response does not really deal with the missing aspect. Approaches being  "dominant approach in industry-level benchmarks" does not mean anything. "superior scalability and flexibility for capturing the intricate relationships within these datasets" is just not true and "build upon an established foundation, ensuring comparability and reproducibility of our results" is a straw man fallacy.
>
> W7: My understanding of that work is that it does both in one step. It creates a graph according to a schema. My point was not so much about comparing with that paper, but more about whether it makes sense to have multiple expert created schema and have the system find out which one is most beneficial.

---

> > ### Author Response · Authors · 2024-11-22
> >
> > Thanks for the further feedback.
> >
> > **NQ1:**
> > We added new experiments on other classification tasks, as shown below:
> >
> > |            | MAG (venue) | MAG (year) | Stackexchange (churn) |
> > |------------|-------------|------------|-----------------------|
> > | Overall    | 47.58       | 51.17      | 85.43                 |
> > | First bin  | 47.57       | 44.79      | 83.69                 |
> > | Second bin | 49.02       | 49.24      | 78.40                 |
> > | Third bin  | 47.18       | 52.40      | 87.10                 |
> > | Fourth bin | 45.71       | 53.91      | 90.69                 |
> > | Fifth bin  | 48.42       | 55.54      | 91.31                 |
> > | DeepFM     | 28.19       | 28.42      | 59.84                 |
> >
> > In general, we observe that different datasets exhibit slightly different patterns, making it challenging to draw universal conclusions. However, one common observation is that nodes with the highest degree tend to benefit most from graph construction.
> >
> > **NQ2:**
> > We acknowledge that modeling relational data with a hypergraph is a valuable research direction. However, there are currently no existing hypergraph frameworks specifically designed for multi-tabular data. While some works have adopted hypergraphs for tabular data [1][2], they have limitations in scope.
> >
> > - For [1], the motivation for using hypergraphs arises primarily from modeling a k-nearest neighbor (kNN)-retrieved graph. Here, the graph structure is implicit, aligning well with hypergraph modeling. Their focus is single table modeling with categorical attributes.
> > - [2] focuses on small tabular datasets, such as LaTeX tables, rather than relational databases.
> >
> > To the best of our knowledge, there is no existing hypergraph architecture explicitly designed for multi-relational graphs, such as relational data that incorporates both intra-table relationships and cross-table primary key–foreign key (PK-FK) relations. While exploring this topic is valuable, it is somewhat beyond the scope of this paper.
> >
> > [1] Du, Kounianhua, et al. "Learning enhanced representation for tabular data via neighborhood propagation." Advances in Neural Information Processing Systems 35 (2022): 16373-16384.
> > [2] Chen, Pei, et al. "HYTREL: Hypergraph-enhanced tabular data representation learning." Advances in Neural Information Processing Systems 36 (2024).
> >
> > **NQ3:**
> > We think that having multiple expert-created schemas and enabling the system to determine the most beneficial one makes sense. The schema design in knowledge graph (KG) construction differs somewhat from graph construction in our case.
> >
> > - In KG, the schema focuses on the quality of relation extraction and whether the graph captures all necessary information.
> > - In our case, both tabular modeling and graph modeling utilize all available information; the key difference lies in *how* the information is used—as features or as relations.
> >
> > In our implementation, the LLM effectively acts as the expert, generating multiple schemas. We then use the model as an anchor to determine which schema performs best.

---

### Official Review · Reviewer_oErT · 2024-11-05

**Soundness:** 3
**Presentation:** 2
**Contribution:** 2
**Rating:** 6
**Confidence:** 4

**Summary:**

This paper proposes a method to convert table data into a graph by leveraging a large language model (LLM).
The primary features are as follows:
- Detailed design of LLM prompts.
- Comprehensive definition of basic actions (Section 4.2).
- Quantitative oracle using GML, potentially effective for this task (Section 4.3).

**Strengths:**

- S1: The proposal introduces a carefully devised LLM prompt (Appendix D)
- S2: The evaluation experiments show that the proposal significantly outperforms existing techniques and achieves results close to those of manual graph generation.
- S3: Section 3.1 identifies C4 (graph variations) , which is indeed an important challenge.
- S4: Generating co-author relationships as edges is definitely effective for node classification of homophily graphs.

**Weaknesses:**

- W1: The problem definition is not explicitly stated.

- W2: Although the proposal utilizes carefully designed LLM prompts, it relies on standard techniques in LLMs like few-shot learning and chain of thought (CoT), making the novelty unclear.

- W3: While the quantitative oracle evaluating GML with a validation set seems effective, Table 4 suggests the oracle may not be essential. The GML results are highly dependent on the choice of label selection (e.g., venue vs. year), making the conclusion "LLMs can generate good candidates merely based on prior knowledge" is not reasonable. We encourage the authors more detailed discussions here.

- W4: Concerning the five challenges in Section 3.1, if the schema is normalized, issues like C2 and C3 (1NF) seem unlikely to arise. Additionally, the proposal only addresses node classification and does not handle tasks like link prediction or node clustering, so C5 appears to be an overstatement.

- W5: There is a lack of evaluation on speed improvements. Section 4.2 claims the high cost of JTD, and Section 4.3 mentions a design for potential speed-ups, making such an evaluation essential.

**Questions:**

Questions
- Is the problem defined as follows?  The input: relational data, GML for downstream tasks, and records and class labels for training and validation data. output: transformed graph data from the relational data.

Comments
- The term "RGAT" may be incorrect and should be "GAT." The cited paper (Veličković et al., 2017) refers to GAT.
- Using GML models suited for homophilic graphs (e.g., RGCN, GAT) as an oracle may be unsuitable for tasks on heterophilic graphs. This issue could be addressed by selecting GML models tailored to heterophilic graphs as the oracle.

---

> ### Author Response · Authors · 2024-11-20
>
> Thanks for your time and efforts in reviewing.
> 1. W1: The problem definition is not explicitly stated.
>
> We are willing to clarify the problem definition here.
>
> *   **Input:** A collection of tables with an unknown or incomplete relational structure [1]; a predictive task defined on the set of tables, along with some observable training labels.
> *   **Output:** A heterogeneous graph designed for this predictive task, which should help GML models achieve good task performance.
> *   **Evaluation:** Quality of the graph, and we adopt GML models' performance as anchors to reflect the graph quality.
>
>
> [1] Gan, Quan, et al. "Graph Machine Learning Meets Multi-Table Relational Data." Proceedings of the 30th ACM SIGKDD Conference on Knowledge Discovery and Data Mining. 2024.
>
> 2. 2. W2: Although the proposal utilizes carefully designed LLM prompts, it relies on standard techniques in LLMs like few-shot learning and chain of thought (CoT), making the novelty unclear.
>
> Thank you for your feedback. We want to emphasize that our primary contribution lies in introducing a novel framework for converting tabular data into graph structures, enabling Large Language Models (LLMs) to perform complex analysis. While we utilize established prompting strategies like chain-of-thought, our focus is on how these techniques can be effectively applied within our framework to unlock the potential of LLMs for tabular data analysis.
>
> Specifically, we observed that directly prompting LLMs for decisions without intermediate reasoning steps often led to the identification of only trivial relationships. By incorporating chain-of-thought prompting, LLMs will investigate the input data by explicitly stating it. We empower the LLMs to extract significantly more complex relationships derived from column statistics.  Our work provides valuable insights into the effective application of chain-of-thought within this context, ultimately advancing the ability of LLMs to analyze and interpret tabular data.
>
> 3. W3: While the quantitative oracle evaluating GML with a validation set seems effective, Table 4 suggests the oracle may not be essential. The GML results are highly dependent on the choice of label selection (e.g., venue vs. year), making the conclusion "LLMs can generate good candidates merely based on prior knowledge" is not reasonable. We encourage the authors more detailed discussions here.
>
> It's true that LLMs often generate effective schemas. However, they can sometimes introduce relationships detrimental to downstream tasks. For instance, when predicting publication years in the MAG dataset, LLMs might link papers based on relationships that introduce heterophily [1]. This can significantly hinder performance, as nodes with different labels become erroneously connected.
>
> Our oracle mechanism acts as a safeguard against such scenarios, ensuring the generated schema is optimized for the specific task. This highlights the necessity of a robust pipeline that combines the generative power of LLMs with the refining capabilities of an oracle to address potential shortcomings and ensure optimal performance.
>
> Moreover, you're correct that the original statement lacked precision. We've revised it in the updated version to more accurately reflect our findings.
>
> Revised Statement (Line 459):  "While we observe that LLMs can often generate effective candidate priors independently,  incorporating an oracle remains necessary in certain cases to prevent bad cases.  Developing a method to pre-determine the necessity of an oracle is a promising avenue for future research that could enhance the efficiency of our framework.
>
> [1] Luan, Sitao, et al. "The heterophilic graph learning handbook: Benchmarks, models, theoretical analysis, applications and challenges." arXiv preprint arXiv:2407.09618 (2024).

---

> > ### Comment · Reviewer_oErT · 2024-11-21
> >
> > The responses below are insightful.
> >
> > > Specifically, we observed that directly prompting LLMs for decisions without intermediate reasoning steps often led to the identification of only trivial relationships. By incorporating chain-of-thought prompting, LLMs will investigate the input data by explicitly stating it. We empower the LLMs to extract significantly more complex relationships derived from column statistics. Our work provides valuable insights into the effective application of chain-of-thought within this context, ultimately advancing the ability of LLMs to analyze and interpret tabular data.
> >
> > > It's true that LLMs often generate effective schemas. However, they can sometimes introduce relationships detrimental to downstream tasks. For instance, when predicting publication years in the MAG dataset, LLMs might link papers based on relationships that introduce heterophily [1]. This can significantly hinder performance, as nodes with different labels become erroneously connected.
> > > Our oracle mechanism acts as a safeguard against such scenarios, ensuring the generated schema is optimized for the specific task. This highlights the necessity of a robust pipeline that combines the generative power of LLMs with the refining capabilities of an oracle to address potential shortcomings and ensure optimal performance.

---

> > > ### Author Response · Authors · 2024-11-23
> > >
> > > Dear Reviewer oErT,
> > > We are very glad that you found our responses in the rebuttal “insightful”. Given that it is approaching the end of the discussion period and you maintained your original negative 5 rating, we are writing to check if our rebuttal clarified your concerns and if you have further concerns after the rebuttal. We are very happy to further discuss with you

---

> > > > ### Comment · Reviewer_oErT · 2024-11-23
> > > >
> > > > Indeed, [2,3] is good evidence to make GCN and GAT effective for heterophily graphs.  I raised the score by 1 point assuming that the authors revise the paper based on the new insight (above) and the following evidence.
> > > >
> > > > >> W8. Using GML models suited for homophilic graphs (e.g., RGCN, GAT) as an oracle may be unsuitable for tasks on heterophilic graphs. This issue could be addressed by selecting GML models tailored to heterophilic graphs as the oracle.
> > > >
> > > > > GCN and GAT are actually strong baselines for heterophilic graphs with proper hyperparameter settings [3]. In this paper, what we strengthen is the importance of not introducing heterophilous relations when constructing graphs. There exists no really effective model-centric solution for learning on heterophilic graphs which works consistently better than traditional GCN [2].
> > > >
> > > > > [2] Luan, Sitao, et al. "The heterophilic graph learning handbook: Benchmarks, models, theoretical analysis, applications and challenges." arXiv preprint arXiv:2407.09618 (2024).
> > > > >
> > > > > [3] Luo, Yuankai, Lei Shi, and Xiao-Ming Wu. "Classic GNNs are Strong Baselines: Reassessing GNNs for Node Classification." arXiv preprint arXiv:2406.08993 (2024).

---

> > > > > ### Author Response · Authors · 2024-11-23
> > > > >
> > > > > Thanks for the support. We will revise the contents accordingly.

---

> ### Author Response · Authors · 2024-11-20
>
> 4. W4: Concerning the five challenges in Section 3.1, if the schema is normalized, issues like C2 and C3 (1NF) seem unlikely to arise. Additionally, the proposal only addresses node classification and does not handle tasks like link prediction or node clustering, so C5 appears to be an overstatement.
>
> That's a good question. After rethinking the five challenges, we find that C3 can be converted to C2 after normalization. However, we don't think the other challenges can be handled by normalization. The reason is that the decision made during the graph construction process is about how to model a column properly. For example, the objective of challenge 2 is to consider whether the relationship induced by this categorical value is beneficial. This decision needs to consider the semantic relationship between this column and the corresponding downstream tasks, which cannot be solved by normalization. (A student has a category like AI, computer network, system, etc. From the normalization perspective, there's no need to create an augmented table. However, from the graph construction perspective, such a student-subject-student relationship is beneficial for predictive tasks. As a result, the objective of normalization and graph construction is different)
>
> For multi-task settings, we consider retrieval tasks like link prediction or recommendation in datasets like MAG and Diginetica. Node clustering is an unsupervised learning task, which may be valuable for future work beyond our benchmark.
>
>
>
> 5. W5: There is a lack of evaluation on speed improvements. Section 4.2 claims the high cost of JTD, and Section 4.3 mentions a design for potential speed-ups, making such an evaluation essential.
>
> Thanks for pointing this out. The original statement in the paper is not correct. It's not fair to compare the running time of JTD and the whole pipeline since the goal is not totally the same (join discovery is just one target in the pipeline). Compared to JTD, LLM-based join discovery presents two advantages:
> * No need to set the manual threshold
> * No need to fine-tune the sentence-bert model
>
> For fine-tuning, it takes 8 hours to train the fine-tuned JTD model on the webtable pairs with 8x A5000 GPUs. For a comparison of inference, we compare JTD with LLM on the join discovery task. In this setting, we remove all known key relationships between tables.
>
> |           | Movielens |
> |-----------|-----------|
> | JTD       | 65.37s    |
> | LLM       | 7.37s     |
>
> From the results, we can see that LLM is much faster in terms of join discovery. However, JTD is still important for real industrial data with a large number of tables and columns (which may go beyond LLM context length). As pointed out by other reviewers, we can actually combine LLM and JTD to deal with a large number of columns. We first filter columns with high similarity and feed them to LLM. In this way, we can combine the advantages of the two methods.
>
>
>
>
> 6. Is the problem defined as follows? The input: relational data, GML for downstream tasks, and records and class labels for training and validation data. output: transformed graph data from the relational data.
>
> Exactly. It should be mentioned that we focus on multi-tabular data with unknown or incomplete relational structures.
>
> 7. The term "RGAT" may be incorrect and should be "GAT." The cited paper (Veličković et al., 2017) refers to GAT.
>
> RGAT is an extension of the original GAT model for the heterogeneous graph, and we adopt the implementation from [1], we have updated the citation in the original paper.
>
> 8. Using GML models suited for homophilic graphs (e.g., RGCN, GAT) as an oracle may be unsuitable for tasks on heterophilic graphs. This issue could be addressed by selecting GML models tailored to heterophilic graphs as the oracle.
>
> GCN and GAT are actually strong baselines for heterophilic graphs with proper hyperparameter settings [3]. In this paper, what we strengthen is the importance of not introducing heterophilous relations when constructing graphs. There exists no really effective model-centric solution for learning on heterophilic graphs which works consistently better than traditional GCN [2].
>
>
> [1] Wang, Minjie, et al. "4DBInfer: A 4D Benchmarking Toolbox for Graph-Centric Predictive Modeling on Relational DBs." Advances in Neural Information Processing Systems 37 (2025).
>
> [2] Luan, Sitao, et al. "The heterophilic graph learning handbook: Benchmarks, models, theoretical analysis, applications and challenges." arXiv preprint arXiv:2407.09618 (2024).
>
> [3] Luo, Yuankai, Lei Shi, and Xiao-Ming Wu. "Classic GNNs are Strong Baselines: Reassessing GNNs for Node Classification." arXiv preprint arXiv:2406.08993 (2024).

---

> ### Author Response · Authors · 2024-11-21
>
> Thanks for the acknowledgment. If there is anything else we can do to further improve your overall assessment of the paper, we will always be open to your suggestions. Please do let us know.

---

### Meta-Review · Area_Chair_HjGw · 2024-12-10

**Metareview:**

The authors propose a model to convert tables into knowledge graphs using LLMs. In doing so, they made a genuine effort to show that previous research is not able to address the challenges they describe. They also convinced the reviewers in the rebuttal of their undertaking. However, this success has been in part established in the rebuttal and I would like to ask the authors to actually incorporate all of the discussions and experiments of the rebuttal in the final version of the paper (e.g., use appendix if you run out of space).

**Additional Comments On Reviewer Discussion:**

There was a fruitful discussion between authors and reviewers, leading to many satisfied recipients (authors as well as reviewers).

---

### Decision · Program_Chairs · 2025-01-22

Accept (Poster)